# ENHANCING GEOMETRIC PERCEPTION IN VLMS VIA TRANSLATOR-GUIDED REINFORCEMENT LEARNING

**Hao Yu**[1,2]    **Shuning Jia**[1]    **Guanghao Li**[1]    **Wenhao Jiang**[2*]    **Chun Yuan**[1*]

[1]Tsinghua University
[2]Guangdong Laboratory of Artificial Intelligence and Digital Economy (SZ)

## ABSTRACT

Vision-language models (VLMs) often struggle with geometric reasoning due to their limited perception of fundamental diagram elements. To tackle this challenge, we introduce GEOPERCEIVE, a benchmark comprising diagram instances paired with domain-specific language (DSL) representations, along with an efficient automatic data generation pipeline. This design enables the isolated evaluation of geometric perception independently from reasoning. To exploit the data provided by GEOPERCEIVE for enhancing the geometric perception capabilities of VLMs, we propose GEODPO, a translator-guided reinforcement learning (RL) framework. GEODPO employs an NL-to-DSL translator, which is trained on synthetic pairs generated by the data engine of GEOPERCEIVE, to bridge natural language and DSL. This translator facilitates the computation of fine-grained, DSL-level scores, which serve as reward signals in reinforcement learning. We assess GEODPO on both in-domain and out-of-domain datasets, spanning tasks in geometric perception as well as downstream reasoning. Experimental results demonstrate that, while supervised fine-tuning (SFT) offers only marginal improvements and may even impair performance in out-of-domain scenarios, GEODPO achieves substantial gains: $+26.5\%$ on in-domain data, $+8.0\%$ on out-of-domain data, and $+39.0\%$ on downstream reasoning tasks. These findings underscore the superior performance and generalization ability of GEODPO over SFT. All codes are released at `https://github.com/Longin-Yu/GeoPerceive` to ensure reproducibility.

## 1    INTRODUCTION

VLMs have made significant strides in multimodal applications (Team; Team et al., 2023; 2024; Bai et al.; Yu et al., 2025a; Zhu et al., 2023; Wei et al., 2025b; Xu et al., 2024; Wang et al., 2025; Yu et al., 2025b; Wei et al., 2025a; Yue et al., 2023), yet geometry problem solving (GPS) remains particularly challenging. This difficulty primarily stems from the need for accurate *geometric perception*: before engaging in any symbolic reasoning, a model must correctly identify and ground geometric primitives—such as points, lines, and circles—and their spatial relations within a diagram. In practice, even state-of-the-art VLMs frequently misinterpret these primitives, for instance, confusing an intersection with a tangency, as illustrated in Table 1. Moreover, widely adopted end-to-end benchmarks (Lu et al., 2023; Zou et al., 2024) tend to conflate perceptual errors with reasoning failures, thereby obscuring the true source of model limitations (Cho et al., 2025; Zhang et al., 2023; Chen et al., 2021; Cao & Xiao, 2022; Ning et al., 2023; Chen et al., 2022; Liang et al., 2023; Xia et al., 2024; Li et al., 2023; Kosoy et al., 2025). To address this, we frame the problem around two questions: *how can geometric perception be measured*, and *how can it be improved*?

To measure geometric perception unambiguously and at scale, a benchmark must provide both a pixel-level diagram and a formal scoring target that serves as the diagram's meta information. We adopt a DSL as this target and require it to be (i) complete (able to reconstruct the original diagram) and (ii) canonical (one representation per diagram), ensuring that a given prediction does not receive different scores under semantically equivalent ground truths. Following this principle, we present GEOPERCEIVE, where every instance pairs a rendered diagram with its DSL program.

---

*Corresponding author.

Table 1: Natural language captions for different models. Green indicates correct content, while red denotes incorrect content.

| Diagram | Model | Caption |
|---|---|---|
|  | GPT-o3 (Achiam et al., 2023) | The diagram shows two intersecting circles, ... , an external point $U$ lies still farther to the right, ... red chords $BN$ and $JZ$, ... |
| | Qwen3-235B (Team, 2025) | The diagram consists of two overlapping circles. The larger circle has a center labeled as $Z$, and the smaller circle is tangent to the larger circle ... |
| | GEODPO (Ours) | The diagram depicts ... and two circles, the larger centered at $E$, intersecting the smaller at point $J$ and $N$. The smaller circle passes through $J$, $N$ and $U$, ... |

GEOPERCEIVE is produced by two engines: a generation engine that samples diverse GEODSL programs, and a solving engine that instantiates each program via gradient-based optimization and renders a pixel-level diagram. This design enables exact program-level evaluation, and yields an automatically generated, complexity-controllable corpus for both evaluation and training at low cost.

Leveraging GEOPERCEIVE, we can synthesize a large corpus for training geometric perception. However, directly fine-tuning on such data is sensitive to permutation-equivalent programs: many valid orderings describe the same diagram, while SFT optimizes a single instance and encourages surface-form patterns rather than invariants. We therefore adopt a reinforcement learning paradigm. A straightforward alternative is to train the VLMs to emit DSL directly. However, this departs from its NL-centric pretraining and thus requires more data and training steps to reach stable performance. To bridge this gap without forcing the model off its NL manifold, our proposed approach GEODPO keeps the VLMs in the NL regime and introduces an NL-to-DSL translator trained on synthetic NL–DSL pairs generated by the GEOPERCEIVE pipeline. The translator assigns fine-grained DSL-level scores to model outputs, which we convert into reward signals for preference-based alignment (Rafailov et al., 2024). This design avoids the order-permutation explosion that hinders sequence-level SFT, mitigates distribution shift from NL pretraining, and yields perception-focused supervision at scale without requiring human-labeled data.

Empirically, the translator achieves over $90\%$ F1 on detecting points, lines, and constraints. Meanwhile, GEODPO improves the primary perception score from $41.0$ to $51.9$ (a relative gain of $+26.5\%$), with consistent improvements across out-of-domain settings: $+8.0\%$ on perception and $+39.0\%$ on downstream reasoning. In contrast, SFT provides only marginal benefits and can even degrade performance under distribution shift.

Our contributions are threefold:

- We introduce GEOPERCEIVE, an automatically generated, complexity-controllable benchmark centered on GEODSL, a canonical and ambiguity-free DSL that assigns a unique program to each diagram. This enables exact program-level evaluation and supports a scalable data pipeline for automatic training data generation, significantly reducing data-related costs.

- We propose GEODPO, a translator-guided preference learning framework in which an NL2DSL translator bridges natural language and GEODSL. The translator's soft scores are converted into reward signals for DPO-based alignment, addressing order-permutation issues and distributional shift while keeping the model aligned with its NL pretraining distribution.

- We conduct extensive experiments demonstrating strong and robust improvements over SFT across diverse settings: $+26.5\%$ in-domain perception, $+8.0\%$ OOD perception, and $+39.0\%$ downstream reasoning. Qualitative analyses further reveal substantially fewer geometric hallucinations.

## 2 RELATED WORK

**Formal Representation of Geometry and GPS Benchmarks.** Recent progress in geometric perception has led to the development of various benchmarks and diagram representation frameworks. Krueger et al. (2021) introduced the Geometry Model Building Language (GMBL) to specify geometry problems and construction metadata; however, its sequential design introduces ambiguity by allowing multiple semantically equivalent representations. AlphaGeometry (Trinh et al., 2024)

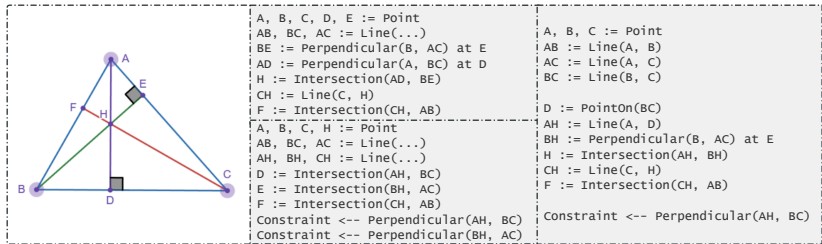

Figure 1: Pseudo-code of existing DSLs' common literals (Trinh et al., 2024; Krueger et al., 2021; Lu et al., 2021). The three DSL programs on the right are semantically equivalent, each describing the same diagram shown on the left. This one-to-many correspondence results in non-unique interpretations, thereby hindering rigorous and consistent evaluation.

proposed a formal language that facilitates symbolic reasoning, but its path-dependent formulation also results in ambiguous descriptions, hindering precise evaluation. Inter-GPS (Lu et al., 2021) designed a predicate-and-parameter-based representation to explicitly capture geometric relationships; nevertheless, its verbose syntax and inconsistent granularity make stable model training difficult. Other related works (Chen et al., 2021; Zhang et al., 2023; Chen et al., 2022; Lu et al., 2023) provide valuable datasets but largely rely on answer-aware metrics, lacking a unified and controllable framework for perception evaluation. Hence, the development of a concise and unambiguous formal language, together with a standardized evaluation system, remains crucial for rigorous assessment of geometric perception.

**Geometry Perception in Vision-Language Models.** Multimodal GPS requires the joint processing of diagrams and textual descriptions (Cho et al., 2025; Zhang et al., 2025). Current VLMs generally fuse diagram features with textual prompts to perform end-to-end reasoning (Cho et al., 2025; Zhang et al., 2023; Chen et al., 2021; Cao & Xiao, 2022; Ning et al., 2023; Chen et al., 2022; Liang et al., 2023; Lu et al., 2023). However, evaluation practices that rely solely on final-answer accuracy often conceal critical geometric perception errors. In particular, existing assessments cannot effectively disentangle errors arising from visual misinterpretation versus textual misunderstanding, thereby obscuring the underlying causes of multimodal hallucinations.

**Reinforcement Learning in GPS.** RL (Schulman et al., 2017; Rafailov et al., 2024; Shao et al., 2024) has been extensively applied in natural language tasks to better align model outputs with task-specific objectives. Recent research has further extended RL to mathematical reasoning, encompassing theorem proving (Kaliszyk et al., 2018), mathematical word problems (Wang et al., 2018; Lu et al., 2022), and algebraic computation (Chen et al., 2018). Within the domain of geometry, GeoDRL (Peng et al., 2023) formulates problem solving as a Markov Decision Process, generating complete solutions directly. In contrast, we leverage RL to guide the translation from NL to DSL, with a particular emphasis on enhancing geometric consistency in perception. Instead of adopting an end-to-end problem-solving framework, our method employs RL to refine intermediate representations, thereby improving both the accuracy and reliability of the generated DSL.

## 3 THE CONSTRUCTION OF GEOPERCEIVE

In this section, we present GEOPERCEIVE, a benchmark specifically designed to evaluate the geometric perception capabilities of VLMs. GEOPERCEIVE focuses exclusively on geometric perception, aiming to assess models' ability to accurately recognize fundamental geometric primitives such as points, lines, and circles. This focused evaluation is crucial for identifying and mitigating geometric hallucinations in VLMs. To overcome the limitations of human-annotated datasets, GEOPERCEIVE adopts an automated pipeline to generate large-scale and diverse training and testing data. The proposed GEOPERCEIVE consists of two key components built upon GeoDSL: (i) a GeoDSL **generation engine** and (ii) a diagram **solving engine**. GeoDSL defines a domain-specific language tailored for geometric perception. The generation engine samples a diverse set of DSL programs, while the solving engine renders each program into pixel-level diagrams. The following sections provide a detailed description of these components.

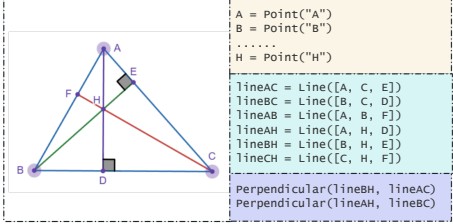

Figure 2: Illustrations of GEODSL syntax. A program may consist of up to four sections: points, lines, circles, and explicit constraints. Point–curve incidence relationships are automatically inferred within curve declarations.

## 3.1 GEODSL: DOMAIN-SPECIFIC LANGUAGE FOR GEOMETRY PERCEPTION

Unlike existing formal languages such as Alpha Geometry (Trinh et al., 2024), Geo-model-builder (Krueger et al., 2021), and Inter-GPS (Lu et al., 2021), which are susceptible to semantic ambiguities (i.e., a single diagram can correspond to multiple DSL programs, as illustrated in Figure 1) and redundant expressions, we propose GEODSL, a concise domain-specific language that establishes a one-to-one correspondence between visual primitives in a diagram and their formal representations, thereby effectively eliminating these issues.

**Design Principles.** The design of GEODSL follows two key principles: **1) Unambiguity.** Each valid geometric relationship corresponds to exactly one GEODSL program, ensuring deterministic evaluation. More specifically, instead of constructive instructions (e.g., "connect $AB$ and $CD$ to form lines $\ell_1$ and $\ell_2$; $\ell_1$ and $\ell_2$ intersect at $E$"), we use descriptive, relational statements (e.g., "points $A, B, E$ lie on $\ell_1$" and "points $C, D, E$ lie on $\ell_2$"). This avoids construction non-uniqueness and yields a canonical, order-invariant set of DSL literals. **2) Simplicity.** The language adopts a minimal set of literals sufficient to describe standard Euclidean constructions, guaranteeing that program length grows linearly with the number of elements, thereby facilitating stable optimization during training.

**Definitions.** Following Lu et al. (2021), we define that: 1) A *primitive* is a basic element in the geometric diagram, i.e., a point, a line, or a circle. 2) A *predicate* is a function that takes some parameters and optionally returns a value. 3) A *literal* is an application of one *predicate* to a sequence of arguments. GEODSL consists of four mutually independent literal categories: point, line, circle, and constraint declarations. Formally, GEODSL regards a geometry construction as a tuple $G = \langle \mathcal{P}, \mathcal{L}, \mathcal{C}, \mathcal{R} \rangle$, in which $\mathcal{P}, \mathcal{L}, \mathcal{C}$ represent a set of `Point`, `Line`, `Circle` literals respectively, and $\mathcal{R}$ represents a set of constraint literals. We call such a tuple $G$ a GEODSL program. Some examples are shown in Figure 2. The details are listed in Appendix A.

## 3.2 BENCHMARK DEFINITIONS AND METRICS

Each instance in GEOPERCEIVE is represented as a tuple $\langle G, D \rangle$, where $G = \langle \mathcal{P}, \mathcal{L}, \mathcal{C}, \mathcal{R} \rangle$ denotes the GEODSL program, and $D \in \mathbb{R}^{3 \times H \times W}$ is the rendered diagram. A model $\mathcal{M}$ under evaluation takes the diagram $D$ as input and predicts a GEODSL program $\hat{G} = \mathcal{M}(D) = \langle \hat{\mathcal{P}}, \hat{\mathcal{L}}, \hat{\mathcal{C}}, \hat{\mathcal{R}} \rangle$. The final score of perception is then calculated by the similarity between $G$ and $\hat{G}$, which is denoted as $\text{Score}(G, \hat{G})$.

To define $\text{Score}(G, \hat{G})$, we independently compare the elements belonging to each predicate category. Specifically, for each category, we consider the pair of multisets $(X, \hat{X}) \in \{(\mathcal{P}, \hat{\mathcal{P}}), (\mathcal{L}, \hat{\mathcal{L}}), (\mathcal{C}, \hat{\mathcal{C}}), (\mathcal{R}, \hat{\mathcal{R}})\}$, where $X = \{x_1, \ldots, x_m\}$ and $\hat{X} = \{\hat{x}_1, \ldots, \hat{x}_n\}$ represent the multisets of ground-truth and predicted primitives, respectively. Here, $m = |X|$ and $n = |\hat{X}|$ denote the number of primitives in each set. For each such pair, we construct a similarity matrix $s \in \mathbb{R}^{m \times n}$, where each entry $s_{ij} = s(x_i, \hat{x}_j) \in [0, 1]$ represents the element-wise matching score between primitive $x_i$ from $X$ and $\hat{x}_j$ from $\hat{X}$. The similarity function $s(a, b)$ is defined recursively, as detailed

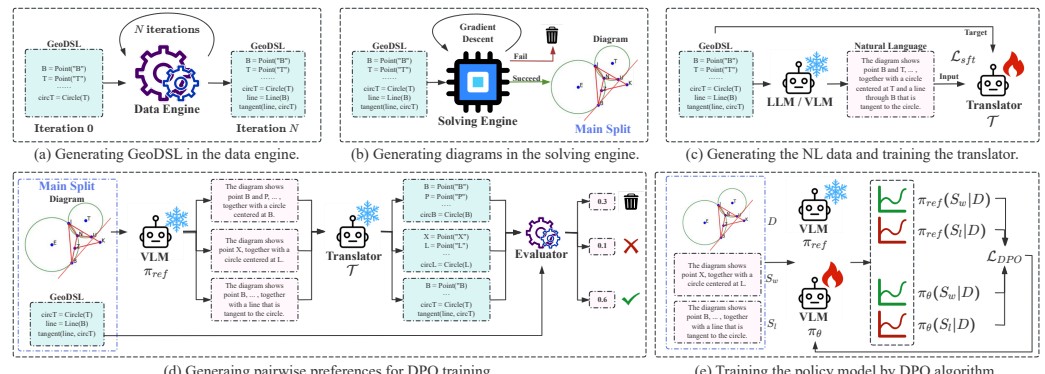

Figure 3: Pipelines of GEOPERCEIVE and GEODPO.

in Equation 1:

$$s(a,b) = \begin{cases} \mathbf{I}[\text{label}(a) = \text{label}(b)] & \text{if } a, b \in \mathcal{P}, \\ F_1(\texttt{P}(a), \texttt{P}(b)) & \text{if } a, b \in \mathcal{L}, \\ \frac{1}{2}\Big[s\big(\texttt{Center}(a), \texttt{Center}(b)\big) + F_1\big(\texttt{P}(a), \texttt{P}(b)\big)\Big] & \text{if } a, b \in \mathcal{C}, \\ \texttt{ConstraintScore}(a, b) & \text{if } a, b \in \mathcal{R}. \end{cases} \quad (1)$$

Here, $\mathbf{I}(\cdot)$ denotes the indicator function, returning 1 if and only if points $a$ and $b$ share the same label, and 0 otherwise. It is assumed that points have unique labels within each GEODSL program (e.g., "A", "B"). For a line or circle $x$, the function $\texttt{P}(x)$ returns the set of labeled points that lie on it. For a circle, $\texttt{Center}(x)$ refers to its labeled center point. The F1-score between two point sets (e.g., $\texttt{P}(a)$ and $\texttt{P}(b)$) is computed recursively by constructing a similarity matrix using label identity, applying the same assignment framework as defined in Equation 1, and computing the F1-score as in Equation 2. The function $\texttt{ConstraintScore}(a, b)$ defines similarity for constraint types, with formal definitions provided in Appendix A.

The optimal one-to-one matching between elements of $X$ and $\hat{X}$ is obtained by solving a maximum-weight bipartite assignment problem, typically via the Hungarian algorithm (Kuhn, 1955) or the Jonker-Volgenant algorithm (Crouse, 2016) (for $m \neq n$). Let $\ell \leq \min(m, n)$ be the number of matched pairs in this assignment, denoted by $(x_{i_k}, \hat{x}_{\pi(i_k)})_{k=1}^{\ell}$. The aggregated similarity for this category is computed as $S(X, \hat{X}) = \sum_{k=1}^{\ell} s_{i_k, \pi(i_k)}$. Based on the aggregated matching score, the per-category *precision*, *recall*, and $F_1$-*score* are defined as:

$$P = \begin{cases} 0 & \text{if } n = 0, \\ \frac{S}{n} & \text{otherwise.} \end{cases} \quad R = \begin{cases} 0 & \text{if } m = 0, \\ \frac{S}{m} & \text{otherwise.} \end{cases} \quad F_1 = \begin{cases} 0 & \text{if } P \cdot R = 0, \\ \left[\frac{1}{2}\left(\frac{1}{P} + \frac{1}{R}\right)\right]^{-1} & \text{otherwise.} \end{cases} \quad (2)$$

To provide a single scalar score for ranking systems, we compute a weighted average of the F1-scores:

$$\text{Score}(G, \hat{G}) = w_P F_1(\mathcal{P}, \hat{\mathcal{P}}) + w_L F_1(\mathcal{L}, \hat{\mathcal{L}}) + w_C F_1(\mathcal{C}, \hat{\mathcal{C}}) + w_R F_1(\mathcal{R}, \hat{\mathcal{R}}). \quad (3)$$

In our experiments, these weights are set equally: $w_P = w_L = w_C = w_R = 0.25$.

### 3.3 GEODSL GENERATION ENGINE

To enable the creation of a large-scale dataset for GEOPERCEIVE, an automated GEODSL generation engine serves as a central component. As illustrated in Figure 3(a), this engine procedurally generates diverse GEODSL programs through a two-stage process:

1. **Primitive Initialization Stage:** The process begins with the random selection of a fundamental geometric configuration. This may involve generating a basic set of primitives, such as a triangle, quadrilateral, or standalone circle, which defines the initial state of the geometric scene.

2. **Iterative Construction Stage:** After initialization, the engine progressively expands the GEODSL program. At each step, it randomly selects a geometric construction operation from a predefined pool. These operations may include adding new points, lines, or circles based

on their relationships with existing elements, or introducing additional constraints. Through this iterative process, increasingly complex geometric configurations are constructed.

A complete list of the construction operations included in the pool is provided in Appendix B.

### 3.4 DIAGRAM SOLVING ENGINE

Once a GEODSL program $G = \langle \mathcal{P}, \mathcal{L}, \mathcal{C}, \mathcal{R} \rangle$ is generated, the next step is to render it as a visual diagram. To this end, we develop a diagram-solving engine that takes the GEODSL program as input and produces the corresponding pixel-based diagram $D$.

While prior work (Krueger et al., 2021) has introduced solving engines, they often rely on different DSLs, which can cause ambiguities when interpreting our GEODSL. In addition, the complexity of their solving processes hampers the integration of new geometric predicates or primitive types, and reported technical issues in certain implementations (e.g., dependence on outdated TensorFlow versions) further restrict their applicability. To overcome these limitations, we develop a new solving engine implemented in PyTorch. This design emphasizes usability, enables straightforward extension with new geometric elements and constraints, and offers a robust and accessible framework for diagram generation.

As shown in Figure 3(b), the core algorithm of our solving engine, described in detail in Appendix C, proceeds through the following key steps:

1. **Primitive Parameterization:** The engine initializes variables corresponding to the geometric parameters of all primitives in $\mathcal{P}$, $\mathcal{L}$, and $\mathcal{C}$. Specifically, each point $P_i \in \mathcal{P}$ is parameterized by its coordinates $(x_i, y_i)$; each line $L_j \in \mathcal{L}$ is parameterized by coefficients $(a_j, b_j, c_j)$ in the implicit form $a_j x + b_j y + c_j = 0$; and each circle $C_k \in \mathcal{C}$ is parameterized by its center $(x_{0k}, y_{0k})$ and radius $r_k$.

2. **Constraint Formulation as Losses:** Every explicit constraint $R_l \in \mathcal{R}$ is converted into a corresponding loss function. For instance, the constraint that a point $P_i(x_i, y_i)$ lies on a line $L_j(a_j, b_j, c_j)$ yields a loss term proportional to $(a_j x_i + b_j y_i + c_j)^2$. Implicit structural constraints, such as requiring that two points defining a line remain collinear with it, are similarly expressed as loss terms.

3. **Visual Plausibility Penalties:** To promote clarity and visual coherence in the rendered diagrams, the objective function incorporates several penalty terms: a density penalty to prevent excessive clustering of points, a distribution penalty to discourage unnatural scattering, and scale and boundary penalties to ensure all elements remain within the canvas at a reasonable size.

4. **Iterative Optimization:** The engine combines all weighted loss and penalty terms into a unified objective function. An iterative optimization algorithm is then applied to determine the primitive parameters that minimize this objective. The process terminates once the total loss falls below a predefined threshold or the maximum number of iterations is reached.

If the solver fails to converge to a satisfactory solution under the given constraints, the corresponding GEODSL instance is labeled as "unsolvable" and excluded from the dataset. Conversely, successfully solved instances yield parameter values that are subsequently used to render the final diagram.

## 4 GEODPO: TRANSLATOR-GUIDED DIRECT PREFERENCE OPTIMIZATION

In this section, we present GEODPO, a translator-guided DPO approach to improve geometric perception. Instead of sequence-level SFT on GEODSL, which is sensitive to permutation-equivalent programs, GEODPO keeps NL outputs, uses an NL-to-DSL translator to score predictions, and turns these scores into preference signals for DPO training. Leveraging GEOPERCEIVE, we synthesize diverse training resources, enabling scalable alignment without human-labeled data.

### 4.1 NL2DSL TRANSLATOR

Training an NL2DSL translator requires a parallel corpus of natural language descriptions paired with their corresponding GEODSL programs. Fortunately, while VLMs often struggle with direct

---

**Algorithm 1** Preference Pair Generation for GEODPO

---

**Require:** Dataset $\mathcal{D}$, Number of samples $N_{samples}$, Minimum score difference threshold $\delta_{min}$, pre-trained model $\pi_{ref}$

1: **function** GENERATEPREFERENCEPAIRS($\mathcal{D}, N_{samples}, \delta_{min}, \pi_{ref}$)
2:      $\mathcal{D}_{DPO} \leftarrow \emptyset$          ▷ Initialize set of DPO training pairs
3:      **for** each $(D, G_{true})$ in $\mathcal{D}$ **do**     ▷ Walk through all pairs of diagram and ground-truth DSL
4:          Let $Samples \leftarrow \emptyset$
5:          **for** $k \leftarrow 1$ to $N_{samples}$ **do**
6:              Sample $S_k \sim \pi_{ref}(D)$        ▷ Sample NL description from the VLM
7:              $s_k \leftarrow \text{Score}(G_{true}, \mathcal{T}(S_k))$        ▷ Score using translator $\mathcal{T}$ and metric
8:              Add $(S_k, s_k)$ to $Samples$
9:          **end for**
10:         $L_{sorted} \leftarrow \text{DescendingSorted}(Samples)$ such that $s_j \geq s_{j+1}$
11:         $idx_w \leftarrow 1$          ▷ Initial index for preferred (winner) sample
12:         $idx_l \leftarrow \lfloor N_{samples}/2 \rfloor + 1$       ▷ Initial index for dispreferred (loser) sample
13:         **while** $idx_w \leq \lfloor N_{samples}/2 \rfloor$ **and** $idx_l \leq N_{samples}$ **do**
14:              $(S_w, s_w) \leftarrow L_{sorted}[idx_w]$
15:              $(S_l, s_l) \leftarrow L_{sorted}[idx_l]$
16:              **if** $s_w - s_l > \delta_{min}$ **then**
17:                  Add $(D, S_w, S_l)$ to $\mathcal{D}_{DPO}$
18:                  $idx_w \leftarrow idx_w + 1$          ▷ Move to the next potential winner
19:                  $idx_l \leftarrow idx_l + 1$       ▷ Move to the next potential loser for the new winner
20:              **else**
21:                  $idx_l \leftarrow idx_l + 1$     ▷ Current loser not distinct enough, try next potential loser
22:              **end if**
23:         **end while**
24:      **end for**
25:      **return** $\mathcal{D}_{DPO}$
26: **end function**

---

NL2DSL translation, they tend to be more adept at the reverse task: generating NL descriptions from structured GEODSL representations. Building on this observation, we construct training data for the translator as illustrated in Figure 3(c). Specifically, we provide existing GEODSL programs as input to a capable VLM and instruct it to generate a descriptive NL paragraph, thereby producing the desired (NL, GEODSL) training pairs.

Using the collected pairs, we fine-tune a language model to serve as the NL2DSL translator $\mathcal{T}$, which takes an NL description $S$ as input and outputs the corresponding GEODSL program $G = \mathcal{T}(S)$. Leveraging the translator $\mathcal{T}$, for a model $\mathcal{M}$, the complete workflow of generating a GEODSL program from a diagram $D$ can be expressed as $G = \mathcal{T}(\mathcal{M}(D))$. The resulting formalized $G$ can then be seamlessly integrated into evaluation and reward signal computation.

### 4.2 TRANSLATOR-GUIDED REINFORCEMENT LEARNING

The key insight is that our translator $\mathcal{T}$, when combined with the GEOPERCEIVE scoring metric (Equation 3), can serve as an effective reward model. This composite system enables us to evaluate the quality of a model-generated NL description $S$ for a given diagram $D$ by translating $S$ into $\hat{G} = \mathcal{T}(S)$ and subsequently scoring $\hat{G}$ against the ground-truth GEODSL program $G_{true}$.

Given a diagram $D$ with its ground-truth GEODSL program $G_{true}$, we prompt the VLM to generate a diverse set of $N_{samples}$ natural language descriptions $\{S_1, S_2, \ldots, S_{N_{samples}}\}$. Each generated description $S_i$ is then evaluated using our reward function, yielding a score $s_i = \text{Score}(G_{true}, \mathcal{T}(S_i))$. Based on these scores, we construct preference pairs $(S_w, S_l)$, where $S_w$ denotes a preferred (winner) description and $S_l$ a dispreferred (loser) description. The procedure for generating these preference pairs is illustrated in Figure 3(d) and detailed in Algorithm 1.

Using the generated preference pairs $(S_w, S_l)$ for each diagram $D$, we fine-tune the VLM with the DPO loss, as illustrated in Figure 3(e). DPO trains a policy $\pi_\theta$ (our VLM) to align with the specified preferences while simultaneously regularizing its deviation from a reference policy $\pi_{ref}$ (typically

the initial state of the VLM prior to DPO fine-tuning). The DPO objective for our task is:

$$\mathcal{L}_{DPO}(\pi_\theta; \pi_{ref}) = -\mathbb{E}_{(D,S_w,S_l)\sim\mathcal{D}_{DPO}} \left[ \log\sigma\left(\beta\log\frac{\pi_\theta(S_w|D)}{\pi_{ref}(S_w|D)} - \beta\log\frac{\pi_\theta(S_l|D)}{\pi_{ref}(S_l|D)}\right)\right]. \quad (4)$$

This loss function drives the model $\pi_\theta$ to assign higher likelihoods to preferred responses $S_w$ and lower likelihoods to dispreferred responses $S_l$, relative to the reference model $\pi_{ref}$. In doing so, it enhances the model's ability to generate high-quality, geometrically accurate natural language descriptions.

## 5 EXPERIMENTS

### 5.1 DATA PREPARATION

The dataset used in our experiments comprises two complementary subsets serving distinct purposes: the *translator* split and the *main* split. Each split contains 10,000 GEODSL samples. The *translator* split pairs NL descriptions with their corresponding GEODSL programs, facilitating the training and evaluation of the NL2DSL Translator. In contrast, the *main* split pairs GEODSL programs with rendered diagrams, enabling the assessment of visual grounding and geometric perception. Detailed statistics of the datasets used in our experiments are provided in Appendix D.

### 5.2 NL2GEODSL TRANSLATOR

**Experimental setup.** The translators in our experiments are initialized from Qwen2.5-7B (Qwen et al., 2024) and trained on the *Translator split* of GEOPERCEIVE. We employ LoRA (Hu et al., 2021) adapters (*rank* = 4) for parameter-efficient fine-tuning. Training is performed for 3 epochs with a batch size of 32, an initial learning rate of $1 \times 10^{-4}$, and a cosine scheduler with linear warm-up over the first 10% of updates. All experiments are conducted on 4 H800 GPUs.

**Results.** Table 2 reports the $F_1$ scores and validity across generation iterations 1 to 5, together with the averaged performance. The translator consistently produces syntactically valid programs and achieves an average $F_1$ above 95% for points and lines. As geometric complexity increases (iterations $4 \sim 5$), performance degrades gracefully, primarily due to the greater difficulty of recovering circular primitives and compound constraints.

Table 2: Element-wise translation accuracy on the GEOPERCEIVE *Translator-test* split. Validity denotes syntactic correctness, while P, R, and $F_1$ correspond to precision, recall, and $F_1$, respectively.

| Iter | Validity | Overall Score | Points | | | Lines | | | Circles | | | Constraints | | |
|---|---|---|---|---|---|---|---|---|---|---|---|---|---|---|
| | | | P | R | F1 | P | R | F1 | P | R | F1 | P | R | F1 |
| 1 | 100.0 | 94.2 | 97.9 | 97.9 | 97.8 | 98.0 | 98.0 | 98.0 | 84.7 | 85.8 | 85.1 | 95.7 | 96.3 | 95.9 |
| 2 | 100.0 | 90.8 | 97.9 | 97.6 | 97.7 | 97.4 | 97.4 | 97.4 | 74.9 | 74.7 | 74.6 | 94.2 | 93.4 | 93.7 |
| 3 | 100.0 | 87.4 | 95.2 | 94.2 | 94.7 | 93.4 | 93.4 | 93.3 | 71.9 | 71.7 | 71.7 | 90.8 | 89.4 | 89.8 |
| 4 | 100.0 | 87.0 | 96.8 | 96.4 | 96.6 | 94.0 | 93.9 | 93.8 | 67.5 | 67.5 | 67.4 | 91.1 | 89.6 | 90.2 |
| 5 | 100.0 | 83.2 | 94.0 | 92.9 | 93.4 | 92.3 | 90.7 | 91.4 | 67.6 | 65.8 | 65.9 | 84.9 | 80.7 | 82.1 |
| **Overall** | 100.0 | 89.2 | 96.4 | 95.9 | 96.1 | 95.4 | 95.0 | 95.2 | 75.0 | 74.8 | 74.6 | 91.7 | 90.4 | 90.8 |

### 5.3 GEODPO

#### 5.3.1 MAIN EXPERIMENTS

**Experimental setup.** We conduct experiments on Qwen2.5-VL (Bai et al., 2025), InternVL3 (Zhu et al., 2025), and LLaVA-Next (Li et al., 2024). For each base model, we consider three settings: (i) the released model without additional supervision, (ii) a model finetuned directly on generated GEODSL, and (iii) a model optimized using our GEODPO method. In GEODPO, the original model is regarded as the reference policy $\pi_{ref}$, with sampling parameters set to $N_{samples} = 10$ and $\delta_{min} = 0.3$, and preference pairs drawn from the *Main-train* split. All finetuning is performed with LoRA adapters (Hu et al., 2021) of rank 8, for one epoch with a batch size of 32, a learning rate of $2 \times 10^{-5}$, and a cosine scheduler with linear warm-up over the first 10% of updates. Unless otherwise specified, training is conducted on 4 H800 GPUs.

**Overall Results.** As shown in Table 3, GEODPO consistently outperforms both the raw models and the SFT ones across three model families. In particular, the overall score improvements range from

Table 3: Scores on the GEOPERCEIVE *Main-test* split.

| Model | Method | Overall Score | Points | | | Lines | | | Circles | | | Constraints | | |
|---|---|---|---|---|---|---|---|---|---|---|---|---|---|---|
| | | | P | R | F1 | P | R | F1 | P | R | F1 | P | R | F1 |
| Qwen2.5-VL (7B) | Raw | 57.96 | 84.58 | 75.19 | 79.07 | 64.15 | 49.73 | 53.95 | 45.16 | 44.82 | 44.63 | 54.30 | 54.12 | 54.18 |
| | w/ SFT | 64.02 (+10.46%) | 83.81 (-0.91%) | 82.18 (+9.3%) | 82.21 (+3.97%) | 64.49 (+0.53%) | 55.74 (+12.09%) | 58.02 (+7.54%) | 58.07 (+28.59%) | 57.61 (+28.54%) | 57.56 (+28.97%) | 58.05 (+6.91%) | 58.76 (+8.57%) | 58.29 (+7.59%) |
| | w/ GeoDPO | 66.19 (+14.2%) | 93.97 (+11.1%) | 85.96 (+14.32%) | 89.26 (+12.89%) | 68.82 (+7.28%) | 60.65 (+21.96%) | 62.3 (+15.48%) | 49.38 (+9.34%) | 48.54 (+8.3%) | 48.7 (+9.12%) | 64.5 (+18.78%) | 64.5 (+19.18%) | 64.5 (+19.05%) |
| InternVL3 (8B) | Raw | 58.44 | 82.07 | 76.09 | 78.59 | 60.52 | 53.47 | 54.61 | 48.70 | 48.02 | 48.16 | 52.43 | 52.37 | 52.40 |
| | w/ SFT | 62.71 (+7.31%) | 85.96 (+4.74%) | 85.35 (+12.17%) | 84.68 (+7.75%) | 62.28 (+2.91%) | 60.12 (+12.44%) | 58.8 (+7.67%) | 58.6 (+20.33%) | 58.83 (+22.51%) | 58.28 (+21.01%) | 49.12 (-6.31%) | 49.16 (-6.13%) | 49.09 (-6.32%) |
| | w/ GeoDPO | 67.41 (+15.35%) | 92.15 (+12.28%) | 84.11 (+10.54%) | 87.44 (+11.26%) | 70.57 (+16.61%) | 56.24 (+5.18%) | 60.58 (+10.93%) | 59.79 (+22.77%) | 58.8 (+22.45%) | 59.12 (+22.76%) | 62.5 (+19.21%) | 62.5 (+19.34%) | 62.5 (+19.27%) |
| LLaVA-Next (7B) | Raw | 41.01 | 59.65 | 46.05 | 50.40 | 46.45 | 31.94 | 36.27 | 35.42 | 36.01 | 35.34 | 42.68 | 41.82 | 42.02 |
| | w/ SFT | 51.1 (+24.6%) | 72.25 (+21.12%) | 69.84 (+51.66%) | 69.94 (+38.77%) | 50.07 (+7.79%) | 50.6 (+58.42%) | 48.31 (+33.2%) | 44.18 (+24.73%) | 44.85 (+24.55%) | 44.19 (+25.04%) | 41.81 (-2.04%) | 42.21 (+0.93%) | 41.96 (-0.14%) |
| | w/ GeoDPO | 51.86 (+26.46%) | 74.35 (+24.64%) | 64.22 (+39.46%) | 66.98 (+32.9%) | 57.87 (+24.59%) | 47.56 (+48.9%) | 51.66 (+42.43%) | 46.8 (+32.13%) | 44.12 (+22.52%) | 42.61 (+20.57%) | 46.18 (+8.2%) | 46.18 (+10.43%) | 46.18 (+9.9%) |

+14.2% to +26.46% relative to the raw models. These results confirm that GEODPO substantially enhances geometric grounding and overall model performance, independent of the model family.

**Element-wise Performance.** Across all perception types (Points, Lines, Circles, Constraints), GEODPO consistently improves upon the raw models. A notable trend emerges in the *Constraints* category: while SFT models exhibit slight performance degradation for models such as InternVL3 and LLaVA-Next, GEODPO effectively alleviates this issue, achieving significant gains. For example, constraint performance increases by +9.9% to +19.27% across models with GEODPO, in contrast to the decline observed in SFT models. This indicates that GEODPO not only strengthens fundamental geometric features (e.g., Points and Lines) but also stabilizes the more fragile aspects of constraint reasoning, which tend to be negatively affected by SFT.

### 5.3.2 OUT-OF-DISTRIBUTION EVALUATION

**Setup.** To assess robustness under distribution shifts, we evaluate (i) *perception* using a hand-crafted GEOPERCEIVE-OOD set consisting of 100 diagrams curated by 10 postgraduate annotators from public sources[1] , with the same evaluation metric as in our main experiments; and (ii) *reasoning* on a 203-item MathVista subset labeled "geometry diagram" and "multi-choice."

Table 4: Scores on the GEOPERCEIVE-OOD.

| Model | Method | Overall Score | Points | | | Lines | | | Circles | | | Constraints | | |
|---|---|---|---|---|---|---|---|---|---|---|---|---|---|---|
| | | | P | R | F1 | P | R | F1 | P | R | F1 | P | R | F1 |
| Qwen2.5-VL (7B) | Raw | 58.14 | 80.42 | 75.11 | 77.67 | 67.53 | 49.74 | 57.29 | 43.82 | 44.63 | 44.22 | 57.26 | 49.95 | 53.36 |
| | w/ SFT | 58.41 (+0.46%) | 80.63 (+0.26%) | 74.15 (-1.28%) | 77.25 (-0.54%) | 67.53 (0.0%) | 50.37 (+1.27%) | 57.7 (+0.72%) | 46.17 (+5.36%) | 45.66 (+2.31%) | 45.92 (+3.84%) | 56.5 (-1.33%) | 49.52 (-0.86%) | 52.78 (-1.09%) |
| | w/ GeoDPO | 60.28 (+3.68%) | 81.89 (+1.83%) | 79.98 (+6.48%) | 80.92 (+4.18%) | 68.19 (+0.98%) | 52.84 (+6.23%) | 59.54 (+3.93%) | 44.42 (+1.37%) | 45.69 (+2.38%) | 45.05 (+1.88%) | 60.49 (+5.64%) | 51.44 (+2.98%) | 55.6 (+4.2%) |
| InternVL3 (8B) | Raw | 58.74 | 76.06 | 76.90 | 76.48 | 63.92 | 51.04 | 56.76 | 51.69 | 50.48 | 51.08 | 49.21 | 52.14 | 50.64 |
| | w/ SFT | 58.57 (-0.29%) | 76.41 (+0.46%) | 77.73 (+1.08%) | 77.06 (+0.76%) | 63.52 (-0.63%) | 51.43 (+0.76%) | 56.84 (+0.14%) | 49.84 (-3.58%) | 48.22 (-4.48%) | 49.02 (-4.03%) | 49.97 (+1.54%) | 52.85 (+1.36%) | 51.37 (+1.44%) |
| | w/ GeoDPO | 60.91 (+3.69%) | 79.37 (+4.35%) | 79.36 (+3.2%) | 79.37 (+3.78%) | 67.53 (+5.65%) | 52.33 (+2.53%) | 58.97 (+3.89%) | 52.92 (+2.38%) | 52.17 (+3.35%) | 52.54 (+2.86%) | 50.71 (+3.05%) | 54.98 (+5.45%) | 52.76 (+4.19%) |
| LLaVA-Next (7B) | Raw | 42.48 | 61.98 | 49.00 | 54.73 | 46.86 | 33.12 | 38.81 | 37.03 | 35.65 | 36.33 | 40.96 | 39.21 | 40.07 |
| | w/ SFT | 43.5 (+2.4%) | 60.95 (-1.66%) | 54.47 (+11.16%) | 57.53 (+5.12%) | 45.79 (-2.28%) | 35.36 (+6.76%) | 39.9 (+2.81%) | 35.62 (-3.81%) | 37.43 (+4.99%) | 36.5 (+0.47%) | 40.96 (0.0%) | 39.23 (+0.05%) | 40.07 (0.0%) |
| | w/ GeoDPO | 45.88 (+8.0%) | 68.35 (+10.28%) | 54.09 (+10.39%) | 60.39 (+10.34%) | 52.37 (+11.76%) | 37.27 (+12.53%) | 43.55 (+12.21%) | 39.79 (+7.45%) | 36.43 (+2.19%) | 38.04 (+4.71%) | 42.25 (+3.15%) | 40.87 (+4.23%) | 41.55 (+3.69%) |

**Results.** On GEOPERCEIVE-OOD (Table 4), SFT proves fragile, often leading to marginal improvements or even regressions, whereas GEODPO consistently enhances overall OOD scores across all backbones, accompanied by concurrent category-wise gains. On MathVista (Table 5), GEODPO yields substantial downstream improvements over both raw and SFT models (e.g., InternVL3: $29.06 \rightarrow 40.39$, corresponding to a gain of $+39.0\%$), while SFT offers only limited benefits.

Table 5: Geometry subset of MathVista.

| Method | Models | | |
|---|---|---|---|
| | Qwen2.5-VL | InternVL3 | LLaVA-Next |
| Raw | 30.05 | 29.06 | 26.11 |
| w/ SFT | 32.02 | 30.05 | 29.06 |
| w/ GeoDPO | **40.39** | **40.39** | **33.50** |

Overall, our experiments demonstrate that trajectory-level RL reward optimization (GEODPO) achieves stronger performance and generalization than token-level log-likelihood SFT, with this advantage becoming particularly pronounced under distribution shifts.

---

[1]Lu et al. (2021), Lu et al. (2023), and https://www.imo-official.org/

## 6 CONCLUSION

We addressed geometric perception in VLMs through two complementary directions. For *measurement*, we introduced GEOPERCEIVE, built upon GEODSL, a canonical and ambiguity-free DSL that assigns each diagram a unique program. This enables exact program-level evaluation and supports an automatically generated, complexity-controllable data pipeline, yielding scalable training data at low cost. For *improvement*, we proposed GEODPO, a translator-guided preference learning framework in which an NL2DSL translator bridges natural language and GEODSL. Its soft scores are transformed into reward signals for DPO-based alignment, thereby mitigating order-permutation issues and distribution shifts while keeping the model close to its NL pretraining manifold. Experiments demonstrate strong and robust gains over SFT: in-domain perception +26.5%, OOD perception +8.0%, and downstream reasoning +39.0%, along with substantially fewer geometric hallucinations.

## ACKNOWLEDGMENT

This work was supported by the National Key R&D Program of China (2022YFB4701400/4701402), SSTIC Grant (KJZD20230923115106012, KJZD20230923114916032, GJHZ20240218113604008).

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

# A   DETAILS OF GEODSL

## A.1   DETAILS OF PREDICATES

Table 6 lists the *element* predicates, each of which returns a primitive geometric object. Table 7 enumerates the *constraint* predicates together with their parameters and semantics.

Table 6: Predicates that return geometric primitives in GeoDSL.

| Predicate | Parameters | Example |
|---|---|---|
| Point | label: str \| null | Point("A"),Point(null) |
| Line | through: list<Point> | Line([Point("A"), Point("B")]) |
| Circle | center: Point \| null
through: list<Point> | Circle(Point("O"), [Point("A")]) |

Table 7: Predicates that return constraint instances in GeoDSL.

| Predicate | Operator | Parameters | Description |
|---|---|---|---|
| ConstraintParallel | $C_\parallel$ | line1:  Line
line2:  Line | Constrain line1 to be parallel to line2. |
| ConstraintPerpendicular | $C_\perp$ | line1:  Line
line2:  Line | Constrain line1 to be perpendicular to line2. |
| ConstraintLineCircleTangent | $T_{LC}$ | line:  Line
circle:  Circle | Constrain line to be tangent to circle. |
| ConstraintCircleCircleTangent | $T_{CC}$ | circle1:  Circle
circle2:  Circle | Constrain circle1 to be tangent to circle2. |
| ConstraintEqual | Eq | dist1:  Tuple<Point, Point>
dist2:  Tuple<Point, Point> | Constrain dist1 and dist2 to have equal length. |

## A.2   IMPLEMENTATION OF CONSTRAINT SCORE

The scoring function `ConstraintScore` introduced in Eq. equation 1 is implemented differently for each constraint type as follows.

$$
\begin{cases}
s\big(C_\parallel(l_1, l_2), C_\parallel(\hat{l}_1, \hat{l}_2)\big) & = F1\Big(\{l_1, l_2\}, \{\hat{l}_1, \hat{l}_2\}\Big), \\
s\big(C_\perp(l_1, l_2), C_\perp(\hat{l}_1, \hat{l}_2)\big) & = F1\Big(\{l_1, l_2\}, \{\hat{l}_1, \hat{l}_2\}\Big), \\
s\big(T_{LC}(l, c), T_{LC}(\hat{l}, \hat{c})\big) & = \frac{1}{2}\Big[s(l, \hat{l}) + s(c, \hat{c})\Big], \\
s\big(T_{CC}(c_1, c_2), T_{CC}(\hat{c}_1, \hat{c}_2)\big) & = F1\Big(\{c_1, c_2\}, \{\hat{c}_1, \hat{c}_2\}\Big). \\
s\big(Eq(d_1, d_2), Eq(\hat{d}_1, \hat{d}_2)\big) & = \frac{1}{2}\max\Big(F1(d_1, \hat{d}_1) + F1(d_2, \hat{d}_2), \ F1(d_1, \hat{d}_2) + F1(d_1, \hat{d}_2)\Big)
\end{cases}
\tag{5}
$$

# B   DETAILS OF GENERATION ENGINE

## B.1   PRIMITIVE INITIALIZATION STAGE

In this stage, we bootstrap the scene with a single, randomly-chosen geometric primitive sampled from a fixed probability distribution. A uniform random number $r \sim \mathcal{U}(0, 1)$ decides whether we insert (i) a triangle (50%), (ii) a quadrilateral (30%), or (iii) a circle (20%). If a circle is selected, we optionally place a random center (70% chance) before creating the circle object. The outcome is exactly one fully-specified primitive, together with all of its constituent points, lines, or arcs, that serves as the seed for subsequent construction stages. The following pseudo code displays the whole process.

```
r ← Uniform(0,1)            # a random value

if r < 0.5 then             # create a triangle with 0.50 probility
    A, B, C ← NewRandomPoints(3)
    AB ← Line([A, B])
    BC ← Line([B, C])
    CA ← Line([C, A])
    AddShape(A, B, C, AB, BC, CA)

elseif r < 0.8 then         # create a quadrilateral with 0.30 probility
    A, B, C, D ← NewRandomPoints(4)
```

```
    AB ← Line([A, B])
    BC ← Line([B, C])
    CD ← Line([C, D])
    DA ← Line([D, A])
    AddShape(A, B, C, D, AB, BC, CD, DA)

else                           # create a cicle with 0.20 probility
    q ← Uniform(0,1)
    if q < 0.7 then
        center ← NewRandomPoint()
        AddShape(center)
    else
        center ← null
    end if
    circle ← Circle(center)
    AddShape(circle)

end if
```

## B.2 ITERATIVE CONSTRUCTION STAGE

### B.2.1 OVERVIEW

After the seed primitive is in place, the generator enters an *open-ended loop* that augments the scene step-by-step until the requested number of extra steps is reached or no further operation is feasible. Every iteration follows the pipeline below:

1. **Enumerate admissible operations.** Each concrete subclass of `ConstructionProcessor` exposes a `prepare()` method that checks whether its pre-conditions can be satisfied on the current scene (e.g., a triangle must already exist before an orthocentre can be drawn, at least two free point labels must remain before sampling two fresh points, etc.). Only the processors whose `prepare()` returns `True` are kept as candidates.

2. **Randomly pick one candidate.** From the candidate set a single processor is drawn uniformly; this adds controlled stochasticity while guaranteeing geometric validity.

3. **Apply the operation.** The chosen processor's `apply()` method instantiates the necessary primitives (points, lines, circles), appends them to the global construction, and registers any incidence or perpendicularity constraints that logically follow. It also returns a structured `ConstructionDescription` that later becomes part of the natural-language instruction sequence.

4. **Record and continue.** The formatted description is appended to the running script; the loop then proceeds to the next iteration.

**Pseudocode overview.** The essence of the loop is captured below; it mirrors the logic inside `generate` and `generate_step` of the accompanying implementation.

```
for i ← 1 ... extra_steps do
    C ← { op in Operations | op.prepare() = True }   # candidates
    if C is empty then
        break                                        # no valid move
    end if

    op ← UniformChoice(C)                            # pick one operation
    op.apply()                                       # apply the selected operation
end for
```

### B.2.2 AVAILABLE OPERATIONS

**Construct Orthocentre.** Given an existing triangle, draw its three altitudes and mark their concurrence point $H$, the orthocentre. Perpendicularity constraints between each altitude and its opposite side are added to the scene.

```
(A, B, C, BC, AC, AB) ← ChooseRandomTriangle()
H ← NewRandomPoint()  with prob. 0.5  else  AnonymousPoint()
AH ← Line(A, H)
BH ← Line(B, H)
CH ← Line(C, H)
AddShape(H, AH, BH, CH)
AddConstraint(Perpendicular, AH, BC)
```

```
AddConstraint(Perpendicular, BH, AC)
AddConstraint(Perpendicular, CH, AB)
```

**Construct Circumcentre.** For a chosen triangle, create its circumcircle through the three vertices and, optionally, generate the centre $O$. No additional constraints are required because the circle already passes through all vertices by construction.

```
(A, B, C, BC, AC, AB) ← ChooseRandomTriangle()
O ← NewRandomPoint()  with prob. 0.5  else  null
circ ← Circle(O, [A, B, C])
AddShape(O, circ)
```

**Construct Incentre.** Insert the incircle of a triangle, enforce tangency to all three sides, and optionally expose the centre $I$ as well as up to three touch points where the circle meets the sides.

```
(A, B, C, BC, AC, AB) ← ChooseRandomTriangle()
I ← NewRandomPoint()  with prob. 0.5  else  null
inc ← Circle(I, [A, B, C])
AddShape(I, inc)
AddConstraint(Tangent, inc, AB)
AddConstraint(Tangent, inc, BC)
AddConstraint(Tangent, inc, AC)
for each side S in {AB, BC, CA} do
    with prob. 0.4:
        P ← Intersection(inc, S)
        AddShape(P)
end for
```

**Construct Segment.** Connect two existing points that are not already joined by a segment lying on the same line; this is the simplest way to incrementally densify the point set.

```
(A, B) ← TwoExistingPointsWithoutSegment()
seg ← Line(A, B)
AddShape(seg)
```

**Construct Two Points and Connect.** Sample one new point on each of two (possibly identical) curves—line or circle—and link them with a segment, thereby coupling previously independent objects.

```
(curve1, curve2) ← ChooseTwoCurves()  # lines and circles are both regarded as curves.
A ← NewPointOn(curve1);  B ← NewPointOn(curve2)
AB ← Line(A, B)
AddShape(A, B, AB)
```

**Construct Point and Connect Existing One.** Place a fresh point on a chosen line or circle that does not yet contain the selected existing point, then connect the two points to form a new segment.

```
(P, curve) ← ChoosePointAndCurve()
A ← NewPointOn(curve)
AP ← Line(A, P)
AddShape(A, AP)
```

## C   DETAILS OF SOLVING ENGINE

The engine formulates every geometric construction as a continuous, constrained-optimization problem. All primitives are endowed with trainable real-valued parameters, and a collection of differentiable loss terms is minimised by gradient descent. The global objective is

$$\mathcal{L} = \sum_{k \in \mathcal{C}} w_k \, \mathcal{L}_k \; + \; \lambda \sum_{m \in \mathcal{P}} v_m \, \mathcal{P}_m, \qquad \lambda > 0, \tag{6}$$

where $\mathcal{C}$ and $\mathcal{P}$ index the *hard geometric constraints* and the *soft regularisation penalties*, respectively, while $w_k$ and $v_m$ are user-specified weights.

## C.1 PARAMETERISATION OF PRIMITIVES

- **Points** $P = (x, y) \in \mathbb{R}^2$ are free variables.
- **Lines** $\ell = (a, b, c)$ obey the implicit equation $a\,x + b\,y + c = 0$ with an *approximate* normalisation $a^2 + b^2 \approx 1$ enforced softly.
- **Circles** $\gamma = (c_x, c_y, r)$ consist of a centre $C = (c_x, c_y)$ and a radius $r > 0$.

## C.2 INTERNAL CONSISTENCY LOSSES

These losses ensure that each primitive remains consistent with the incidence information stored inside it.

**Point–Line Incidence.** If a point $P$ is declared to lie on a line $\ell$, the squared (algebraic) distance

$$\mathcal{L}_{\text{inc}}(P, \ell) = \big(a\,x_P + b\,y_P + c\big)^2 \tag{7}$$

is accumulated.

**Line Normalisation.** To avoid degenerate parameterisations we penalise

$$\mathcal{L}_{\text{norm}}(\ell) = \big(\sqrt{a^2 + b^2} - 1\big)^2. \tag{8}$$

**Point–Circle Incidence.** For a circle $\gamma$ and one of its defining points $P$,

$$\mathcal{L}_{\text{on-circle}}(P, \gamma) = \big(\|P - C\| - r\big)^2. \tag{9}$$

**Fixed Centre (optional).** If a circle is required to coincide with a reference centre $C_0$,

$$\mathcal{L}_{\text{centre}}(\gamma) = \|C - C_0\|^2. \tag{10}$$

## C.3 EXTERNAL GEOMETRIC CONSTRAINTS

**Equal Segment Lengths.** For two segments $P_1 P_2$ and $P_3 P_4$,

$$\mathcal{L}_=(P_1, P_2, P_3, P_4) = \big(\|P_1 - P_2\| - \|P_3 - P_4\|\big)^2. \tag{11}$$

**Perpendicular Lines.** Given $\ell_1 = (a_1, b_1, c_1)$ and $\ell_2 = (a_2, b_2, c_2)$,

$$\mathcal{L}_\perp(\ell_1, \ell_2) = \big(a_1 a_2 + b_1 b_2\big)^2. \tag{12}$$

**Parallel Lines.**

$$\mathcal{L}_\|(\ell_1, \ell_2) = \big(a_1 b_2 - a_2 b_1\big)^2. \tag{13}$$

**Line–Circle Tangency.** For a line $\ell = (a, b, c)$ and a circle $\gamma = (C, r)$,

$$\mathcal{L}_{\text{tan}}(\ell, \gamma) = \Big(\frac{|a\,c_x + b\,c_y + c|}{\sqrt{a^2 + b^2}} - r\Big)^2. \tag{14}$$

**Prescribed Length.** Enforcing $\|P_1 - P_2\| = L$:

$$\mathcal{L}_{\text{len}}(P_1, P_2; L) = \big(\|P_1 - P_2\| - L\big)^2. \tag{15}$$

**Prescribed Angle.** Let $\angle P_1 V P_2$ be required to equal $\theta$ (in radians):

$$\mathcal{L}_\angle(V, P_1, P_2; \theta) = \Big(\arccos\frac{(P_1 - V)\cdot(P_2 - V)}{\|P_1 - V\|\,\|P_2 - V\|} - \theta\Big)^2. \tag{16}$$

## C.4 SOFT REGULARISATION PENALTIES

**Point Density.** For every unordered pair of points $(P_i, P_j)$ with distance $d_{ij} = \|P_i - P_j\|$ and a threshold $\tau$,

$$\mathcal{P}_{\text{dens}}(P_i, P_j; \tau) = \begin{cases} \dfrac{1}{d_{ij}^2} - \dfrac{1}{\tau^2}, & d_{ij} < \tau, \\ 0, & d_{ij} \geq \tau. \end{cases} \tag{17}$$

**Global Spread.** Let $\mu = \frac{1}{N} \sum_{i=1}^{N} P_i$ be the centroid of all points. A tolerance radius $\rho$ yields

$$\mathcal{P}_{\text{spread}}(\rho) = \sum_{i=1}^{N} \max\big(0, \|P_i - \mu\| - \rho\big)^2. \tag{18}$$

## C.5 Optimisation Procedure

All free parameters are updated by Adam with a step-decayed learning rate. Gradients are back-propagated through the scalar objective equation 6. Optionally, the solver halts early when *every* individual constraint term $\mathcal{L}_k$ becomes smaller than a user-defined threshold, guaranteeing that hard geometric relations are satisfied to the desired precision.

## C.6 Failure Samples of Solving Engine

This section presents examples of failure cases encountered by our solving engine. As illustrated in Figure 4, these failures typically arise from issues such as high complexity or unsolvability of the given problem.

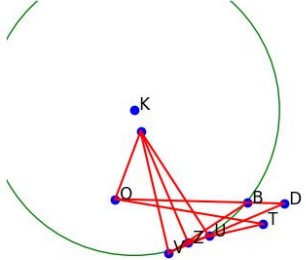 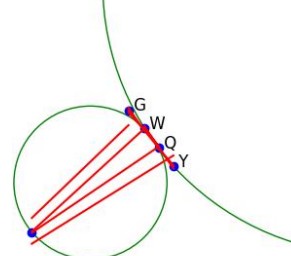

Figure 4: Failure Cases of Solving Engine.

The GEODSLs corresponding to the example images are as follows.

```
# the left case

J = point(label="J")
W = point(label="W")
E = point(label="E")
O = point(label="O")

line_1 = line(through=[J, W])
line_2 = line(through=[W, E])
line_3 = line(through=[E, J, O])

circle_1 = circle(through=[J, W, E, O])

tangent_line_circle(line_2, circle_1)
tangent_line_circle(line_3, circle_1)
tangent_line_circleU = point(label="U")
Z = point(label="Z")
V = point(label="V")
B = point(label="B")
D = point(label="D")
T = point(label="T")
O = point(label="O")
unlabeled_point_1 = point()
K = point(label="K")

line_1 = line(through=[U, Z, T])
line_2 = line(through=[Z, V, B])
line_3 = line(through=[V, U, D])
line_4 = line(through=[B, D, O])
line_5 = line(through=[T, O])
line_6 = line(through=[U, unlabeled_point_1])
line_7 = line(through=[Z, unlabeled_point_1])
line_8 = line(through=[V, unlabeled_point_1])
line_9 = line(through=[O, unlabeled_point_1])

circle_1 = circle(center=K, through=[U, Z, V])

perpendicular(line_6, line_2)
perpendicular(line_7, line_3)
perpendicular(line_8, line_1)
```

```
# the right case

Q = point(label="Q")
```

```
W = point(label="W")
Y = point(label="Y")
unlabeled_point_1 = point()
G = point(label="G")

line_1 = line(through=[Q, W])
line_2 = line(through=[W, Y])
line_3 = line(through=[Y, Q])
line_4 = line(through=[Q, unlabeled_point_1])
line_5 = line(through=[W, unlabeled_point_1])
line_6 = line(through=[Y, unlabeled_point_1])
line_7 = line(through=[Q, G])
line_8 = line(through=[W, G])
line_9 = line(through=[unlabeled_point_1, G])
line_10 = line(through=[Y, G])

circle_1 = circle(through=[Q, W, unlabeled_point_1])
circle_2 = circle(through=[Q, W, Y])

perpendicular(line_4, line_2)
perpendicular(line_5, line_3)
perpendicular(line_6, line_1)
perpendicular(line_7, line_5)
perpendicular(line_8, line_4)
perpendicular(line_9, line_1)
```

## D  DATASET STATISTICS IN MAIN EXPERIMENTS

As described in Section 5, the dataset used in our experiments consists of two complementary subsets with distinct purposes: the *translator* split and the *main* split. The statistics are shown in Table 8. To ensure the Translator receives a balanced and learnable distribution, the *translator* split is sampled using iteration-dependent weights that mitigate the long-tail distribution of more complex programs. For the *main* split, we aim for approximately 2,000 examples per iteration stage. However, the final rendered count decreases at higher complexity levels, as fewer advanced programs can be successfully solved and rendered.

Table 8: Subsets of GEOPERCEIVE in our experiments. "Solving SR" (%) denotes the proportion of GEODSL programs that can be successfully rendered into valid diagrams.

| Generation Iterations | Translator Split | | | | | Main Split | | | | |
|---|---|---|---|---|---|---|---|---|---|---|
| | 1 | 2 | 3 | 4 | 5 | 1 | 2 | 3 | 4 | 5 |
| #Train | 3920 | 2548 | 1960 | 882 | 490 | 1524 | 1485 | 1333 | 1203 | 1034 |
| #Test | 80 | 52 | 40 | 18 | 10 | 47 | 38 | 44 | 38 | 33 |
| #Total | 4000 | 2600 | 2000 | 900 | 500 | 1571 | 1523 | 1377 | 1241 | 1067 |
| Solving SR (%) | – | – | – | – | – | 78.5 | 76.2 | 68.8 | 62.0 | 53.4 |

It is important to note that our synthesis pipeline supports unlimited data generation with controllable complexity, which is different to other benchmarks as shown in Table 9. The statistics reported in this section reflect only the portion of data used in our experiments, not the upper limit of what GEOPERCEIVE can produce. Representative samples from the *main* split are shown in Figure 5.

Table 9: Comparison with existing benchmarks. Unlike reasoning-focused datasets with limited samples, GeoPerceive focuses on perception with infinite automated data generation.

| Benchmark | Domain | Evaluation Granularity | Automation | #Samples |
|---|---|---|---|---|
| Geometry3K (Lu et al., 2021) | Reasoning | Binary Answers | ✗ | 3,002 |
| UniGeo (Chen et al., 2022) | Reasoning | Binary Answers | ✗ | 9,543 |
| MathVista (Lu et al., 2023) | Reasoning | Binary Answers | ✗ | 6,164 |
| GeoQA (Chen et al., 2021) | Reasoning | Binary Answers | ✗ | 4,998 |
| **GEOPERCEIVE (Ours)** | **Perception** | **Primitives + Constraints** | ✓ | ∞ |

## E  MORE EXPERIMENTS

### E.1  EXPLORATION OF ALTERNATIVE RL ALGORITHMS

It is important to note that DPO is not a competing approach, but rather an implementation choice within our proposed RL framework. We selected DPO for its simplicity and direct applicability to the problem at hand. However, our framework allows for the use of alternative RL algorithms.

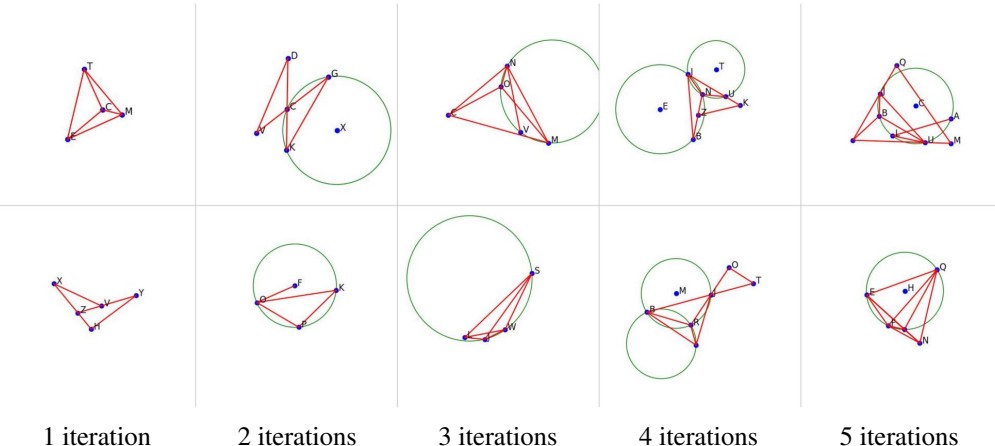

|  |  |  |  |  |
|---|---|---|---|---|
| 1 iteration | 2 iterations | 3 iterations | 4 iterations | 5 iterations |

Figure 5: Randomly selected GEOPERCEIVE diagrams. For each generation iteration, two sampled figures are displayed to illustrate the progressive increase in geometric complexity.

To explore this, we conducted additional experiments using PPO, which involves training both a value model and a policy model. While we found that PPO tends to be slower due to the additional value model, it performs on par with DPO in our framework (as shown in Table 10). This highlights the flexibility of our framework in supporting different RL implementations.

Table 10: Performance of different methods across various metrics.

| Method | Overall | Points | | | Lines | | | Circles | | | Constraints | | |
|---|---|---|---|---|---|---|---|---|---|---|---|---|---|
| | | P | R | F1 | P | R | F1 | P | R | F1 | P | R | F1 |
| Pretrained | 57.96 | 84.58 | 75.19 | 79.07 | 64.15 | 49.73 | 53.95 | 45.16 | 44.82 | 44.63 | 54.30 | 54.12 | 54.18 |
| w/ SFT | 64.02 | 83.81 | 82.18 | 82.21 | 64.49 | 55.74 | 58.02 | 58.07 | **57.61** | 57.56 | 58.05 | 58.76 | 58.29 |
| w/ GeoDPO | **66.19** | **93.97** | 85.96 | **89.26** | 68.82 | **60.65** | **62.30** | 49.38 | 48.54 | 48.70 | **64.50** | **64.50** | **64.50** |
| w/ GeoPPO | 65.54 | 91.13 | **86.18** | 87.89 | **69.05** | 59.81 | 61.58 | **56.51** | 55.70 | **55.66** | 57.12 | 56.98 | 57.04 |

## E.2 ABLATION STUDY ON TRANSLATOR QUILITY

To isolate the effect of the NL2DSL translator from the DPO reinforcement learning signal, we conduct an ablation study in which we train a weaker translator, using less data, and thus achieving lower performance. As shown in Table 11 and Table 12, the performance of the GeoDPO framework significantly degrades when the quality of the translator is compromised. This confirms that the quality of the translator directly influences the effectiveness of the reward signal in our method.

Table 11: Performance of Translator Trained on Different Amounts of Data.

| Training Samples | Validity | Overall | Points | | | Lines | | | Circles | | | Constraints | | |
|---|---|---|---|---|---|---|---|---|---|---|---|---|---|---|
| | | | P | R | F1 | P | R | F1 | P | R | F1 | P | R | F1 |
| 9800 (in practice) | 100.0 | 89.2 | 96.4 | 95.9 | 96.1 | 95.4 | 95.0 | 95.2 | 75.0 | 74.8 | 74.6 | 91.7 | 90.4 | 90.8 |
| 7500 | 99.5 | 89.1 | 94.9 | 94.7 | 94.8 | 94.3 | 94.2 | 94.2 | 76.7 | 75.9 | 76.2 | 91.9 | 91.0 | 91.3 |
| 5000 | 97.5 | 82.3 | 94.5 | 86.8 | 93.7 | 91.7 | 91.6 | 90.1 | 73.9 | 73.1 | 60.4 | 71.6 | 70.3 | 85.0 |
| 2500 | 95.5 | 73.8 | 92.9 | 80.4 | 93.7 | 90.5 | 83.2 | 84.6 | 66.0 | 65.6 | 56.7 | 57.5 | 57.3 | 60.3 |

## E.3 QUALITATIVE ANALYSIS OF PERCEPTION PERFORMANCE

To qualitatively illustrate the performance gap between the pretrained baseline and our GEODPO, we present a representative case study. As shown in Figure 6 and the corresponding prediction logs shown in the following boxes, the pretrained model fails to correctly perceive nearly all geometric relations, resulting in severe hallucinations (marked in red). In contrast, the model trained with GEODPO accurately grounds the majority of visual elements, including the incidence of points on the circumference and the arrangement of chords. One minor error persists: the triangle $EFG$ is misidentified, likely due to the close proximity of point $G$ to the chord $FY$, which creates visual ambiguity.

Table 12: Performance of GeoDPO with Rewards from Translators of Varying Quality.

| Method | Training Samples of Translator | Overall | Points | | | Lines | | | Circles | | | Constraints | | |
|---|---|---|---|---|---|---|---|---|---|---|---|---|---|---|
| | | | P | R | F1 | P | R | F1 | P | R | F1 | P | R | F1 |
| Pretrained | N/A | 57.96 | 84.58 | 75.19 | 79.07 | 64.15 | 49.73 | 53.95 | 45.16 | 44.82 | 44.63 | 54.30 | 54.12 | 54.18 |
| w/ GeoDPO | 9800 (ours) | 66.19 | 93.97 | 85.96 | 89.26 | 68.82 | 60.65 | 62.30 | 49.38 | 48.54 | 48.70 | 64.50 | 64.50 | 64.50 |
| w/ GeoDPO | 7500 | 66.27 | 93.08 | 84.48 | 86.61 | 62.39 | 60.01 | 61.35 | 58.71 | 56.77 | 57.67 | 60.15 | 58.94 | 59.45 |
| w/ GeoDPO | 5000 | 65.48 | 88.70 | 82.01 | 84.53 | 63.12 | 59.42 | 61.04 | 56.98 | 56.17 | 56.29 | 60.08 | 60.08 | 60.08 |
| w/ GeoDPO | 2500 | 63.95 | 85.32 | 79.68 | 82.09 | 61.89 | 56.21 | 59.23 | 57.22 | 54.42 | 56.87 | 58.73 | 55.99 | 57.61 |

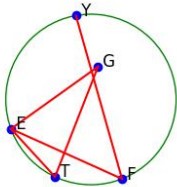

Figure 6: Example Diagram for Perception Performance Difference.

```
# Perception of the Diagram before Training

The diagram depicts a cyclic quadrilateral EFGY inscribed in a circle. The construction
process involves first drawing a circle and then marking points E, F, G, and Y on the
circumference such that the segments EG, GF, FY, and YE are chords of the circle. The
quadrilateral EFGY is then formed by connecting these points in sequence.
```

```
# Perception of the Diagram after Training

The given diagram appears to be constructed by first drawing a circle. Points E, F, T,
and Y are then marked on the circumference of this circle. The red lines connect these
points, forming a triangle EFG and ETG. In addition, there is an extra chord FY
within the circle.
```

## F    LIMITATIONS

This study concentrates on qualitative, diagram–level relationships in geometric problem solving. Purely algebraic constraints receive far less attention. In particular, we do not yet handle specifications such as forcing one segment to be an integer multiple of another, fixing an angle to a prescribed measure, or constraining the area of a polygon (e.g., a triangle) to a given value. Extending our framework to integrate such quantitative requirements remains an important direction for future work.

## G    LLM USAGE STATEMENT

Large language models (LLMs) are used solely for improving grammar and concision. The LLM do not contribute to the conception of ideas, methodological design, experiment execution, analysis or interpretation of results, or the formulation of conclusions. All scientific contributions, insights, and conclusions are made by us.

