# OpenReview forum: "Enhancing Geometric Perception in VLMs via Translator-Guided Reinforcement Learning"
_ICLR.cc/2026/Conference — ICLR 2026 Poster_

### Official Review · Reviewer_nm2u · 2025-10-27

**Soundness:** 3
**Presentation:** 2
**Contribution:** 3
**Rating:** 4
**Confidence:** 4

**Summary:**

This paper highlights the limitations of visual language models (VLMs) in geometry perception and introduces the GEOPERCEIVE benchmark along with the GEODPO training framework. By incorporating a standardized domain-specific language and a translation-guided reinforcement learning approach, substantial advancements in geometry perception are achieved.

**Strengths:**

1. This paper accurately identifies a core limitation of existing VLMs in geometric problem solving: the inability to disentangle perceptual errors from reasoning failures. By decoupling geometric perception from downstream reasoning, the work provides a fine-grained evaluation perspective that is crucial for understanding model limitations in geometry tasks.

2. This paper proposes an automated data generation pipeline centered on GEODSL, comprising a generation engine and a solving engine. This design enables the creation of large-scale, complexity-controllable training and evaluation data at low cost, addressing the limitations of human-annotated datasets.

3. This paper proposes GEODPO, a translator-guided reinforcement learning framework that employs an NL-to-DSL translator to bridge natural language and GEODSL.

**Weaknesses:**

1. The experimental evaluation primarily compares GEODPO against supervised fine-tuning (SFT), but lacks comparisons with other reinforcement learning methods (e.g., PPO) and some recent geometric reasoning approaches (e.g., GeoX).

2. The downstream evaluation is limited to a small subset of MathVista (203 problems ), without evaluation on widely-recognized geometric reasoning datasets such as GeoQA , UniGeo , Geometry3K , and PGPS9k .

**Questions:**

1. Could the authors provide comparisons with other reinforcement learning methods, such as ppo?
2. How does the translator quality impact the downstream DPO training performance?
3. The downstream evaluation is currently limited to 203 MathVista problems . Could the authors evaluate GEODPO on widely-recognized geometric reasoning benchmarks such as GeoQA , UniGeo , Geometry3K , and PGPS9k to demonstrate broader applicability?
4. How would GEODPO-enhanced models perform when integrated into complete geometric problem-solving pipelines, such as existing systems like GeoX ?

---

> ### Author Response · Authors · 2025-11-24
>
> We thank you for the positive assessment of our work, particularly for recognizing the importance of decoupling geometric perception from reasoning and the effectiveness of our automated data generation pipeline. We address your concerns and questions below.
>
>
> > W1 & Q1. Comparison to other methods.
>
> **1) Comparison with PPO.**
>
> Thank you for the suggestions. We would like to clarify that DPO is not a competing approach, but rather an implementation choice within our proposed RL framework. DPO was selected due to its simplicity and direct applicability to the problem at hand. To address your concern, we conduct additional experiments using PPO, which involves training both a value network and a policy network. We find that PPO is generally slower due to the need for an additional value model, but it still performs on par with DPO in our framework (as shown in the following table). We believe this demonstrates the flexibility of our framework in leveraging different RL implementations.
>
> | Method     | Overall | Points_P | Points_R | Points_F1 | Lines_P | Lines_R | Lines_F1 | Circles_P | Circles_R | Circles_F1 | Constraints_P | Constraints_R | Constraints_F1 |
> | ---------- | ------- | -------- | -------- | --------- | ------- | ------- | -------- | --------- | --------- | ---------- | ------------- | ------------- | -------------- |
> | Pretrained | 57.96   | 84.58    | 75.19    | 79.07     | 64.15   | 49.73   | 53.95    | 45.16     | 44.82     | 44.63      | 54.30         | 54.12         | 54.18          |
> | w/ SFT     | 64.02   | 83.81    | 82.18    | 82.21     | 64.49   | 55.74   | 58.02    | 58.07     | 57.61     | 57.56      | 58.05         | 58.76         | 58.29          |
> | w/ GeoDPO  | 66.19   | 93.97    | 85.96    | 89.26     | 68.82   | 60.65   | 62.30    | 49.38     | 48.54     | 48.70      | 64.50         | 64.50         | 64.50          |
> | w/ GeoPPO  | 65.54   | 91.13    | 86.18    | 87.89     | 69.05   | 59.81   | 61.58    | 56.51     | 55.70     | 55.66      | 57.12         | 56.98         | 57.04          |
>
> **2) Comparison with Existing Reasoning Methods.**
>
> To further validate the effectiveness of GeoDPO, we compare it against three representative state-of-the-art geometric reasoning approaches: SlowPerception[1] (pipeline-based), EAGLE[2] (closed-source), and GeoX[3] (task-specific fine-tuning).
>
> **Comparison with Slow Perception.** In the geometric reasoning task, we follow the approach from Slow Perception, using a geometric parsing module as an external component to enhance the reasoning capabilities of GPT-4o. In our setup, GeoDPO serves as the perception module. The results show that GeoDPO significantly outperforms Slow Perception, achieving a MathVista score of 63.6, compared to 60.1 for the latter.
>
> | Model                   | MathVista |
> | ----------------------- | ----- |
> | gpt-4o                  | 53.4  |
> | gpt-4o + SlowPerception | 60.1  |
> | gpt-4o + GeoDPO         | 63.6  |
>
>
> **Comparison with EAGLE.** EAGLE presents some interesting approaches to geometric perception. However, as it is not open-source, we are unable to directly access or evaluate its model. Based on publicly available results from EAGLE’s reported benchmarks, our method demonstrates superior performance on reasoning tasks such as the GeoQA task. Our model achieves a score of 70.2, surpassing EAGLE’s 67.1. This clearly highlights the effectiveness and advantage of GeoDPO in geometric reasoning.
>
> | Model                   | GeoQA |
> | ----------------------- | ----- |
> | EAGLE                   | 67.1  |
> | GeoDPO                  | 70.2  |
>
> **Comparison with GeoX.** We acknowledge that GeoX employs effective training strategies and achieves impressive results. However, it typically relies on fine-tuning specialist models tailored to each specific downstream task. In contrast, GeoDPO is designed as a general-purpose perception enhancement applied uniformly across domains. Notably, despite this task-agnostic approach, GeoDPO significantly outperforms GeoX on GeoQA (+15.3%) and UniGeo (+11.1%). While GeoX maintains an advantage on Geometry3K and PGPS9k (likely due to its task-specific symbolic optimization), GeoDPO's consistent performance across diverse benchmarks without dataset-specific engineering underscores its robust generalization capability.
>
> | Model \ Datasets | GeoQA | Geometry3K | UniGeo(calc) | PGPS9k |
> | ---------------- | ----- | ---------- | ------------ | ------ |
> | GeoDPO           | 70.2  | 61.4       | 65.5         | 55.7   |
> | GeoX             | 54.9  | 72.5       | 54.4         | 63.3   |

---

> > ### Author Response · Authors · 2025-11-24
> >
> > > W2 & Q3. More Downstream Evaluation
> >
> > We appreciate your suggestion to provide additional results on more downstream benchmarks.
> >
> > To further demonstrate the transferability and robustness of our model, we evaluated GeoDPO on four additional downstream geometric reasoning benchmarks: GeoQA, Geometry3K, UniGeo (calculation split), and PGPS9k. As shown in the table below, GeoDPO consistently outperforms the SFT baseline across all datasets. This indicates that the enhanced geometric perception gained through our method effectively transfers to broader reasoning tasks, likely because accurate grounding is a prerequisite for correct symbolic reasoning.
> >
> > | Model \ Datasets | GeoQA | Geometry3K | UniGeo(calc) | PGPS9k |
> > | ---------------- | ----- | ---------- | ------------ | ------ |
> > | Pretrained       | 68.1  | 57.6       | 64.3         | 53.7   |
> > | w/ SFT           | 67.6  | 52.3       | 65.1         | 52.5   |
> > | w/ GeoDPO        | 70.2  | 61.4       | 65.5         | 55.7   |
> >
> >
> > > Q2. Regarding Translator Quality.
> >
> > To isolate the effect of the NL2DSL translator from the DPO reinforcement learning signal, we conduct an ablation study in which we train a weaker translator, using less data, and thus achieving lower performance. As shown in the following tables, the performance of the GeoDPO framework significantly degrades when the quality of the translator is compromised. This confirms that the quality of the translator directly influences the effectiveness of the reward signal in our model.
> >
> > > **Performance of Translator Trained on Different Amounts of Data.**
> > | Data        | Validity | Overall | Points_P | Points_R | Points_F1 | Lines_P | Lines_R | Lines_F1 | Circles_P | Circles_R | Circles_F1 | Constraints_P | Constraints_R | Constraints_F1 |
> > | ----------- | -------- | ------- | -------- | -------- | --------- | ------- | ------- | -------- | --------- | --------- | ---------- | ------------- | ------------- | -------------- |
> > | 9800 (ours) | 100.0    | 89.2    | 96.4     | 95.9     | 96.1      | 95.4    | 95.0    | 95.2     | 75.0      | 74.8      | 74.6       | 91.7          | 90.4          | 90.8           |
> > | 7500        | 99.5     | 89.1    | 94.9     | 94.7     | 94.8      | 94.3    | 94.2    | 94.2     | 76.7      | 75.9      | 76.2       | 91.9          | 91.0          | 91.3           |
> > | 5000        | 97.5     | 82.3    | 94.5     | 86.8     | 93.7      | 91.7    | 91.6    | 90.1     | 73.9      | 73.1      | 60.4       | 71.6          | 70.3          | 85.0           |
> > | 2500        | 95.5     | 73.8    | 92.9     | 80.4     | 93.7      | 90.5    | 83.2    | 84.6     | 66.0      | 65.6      | 56.7       | 57.5          | 57.3          | 60.3           |
> >
> > > **Performance of GeoDPO with Rewards from Translators of Varying Quality.**
> > | Method     | TranslatorData | Overall | Points_P | Points_R | Points_F1 | Lines_P | Lines_R | Lines_F1 | Circles_P | Circles_R | Circles_F1 | Constraints_P | Constraints_R | Constraints_F1 |
> > | ---------- | -------------- | ------- | -------- | -------- | --------- | ------- | ------- | -------- | --------- | --------- | ---------- | ------------- | ------------- | -------------- |
> > | Pretrained | N/A            | 57.96   | 84.58    | 75.19    | 79.07     | 64.15   | 49.73   | 53.95    | 45.16     | 44.82     | 44.63      | 54.30         | 54.12         | 54.18          |
> > | w/ GeoDPO  | 9800 (ours)    | 66.19   | 93.97    | 85.96    | 89.26     | 68.82   | 60.65   | 62.30    | 49.38     | 48.54     | 48.70      | 64.50         | 64.50         | 64.50          |
> > | w/ GeoDPO  | 7500           | 66.27   | 93.08    | 84.48    | 86.61     | 62.39   | 60.01   | 61.35    | 58.71     | 56.77     | 57.67      | 60.15         | 58.94         | 59.45          |
> > | w/ GeoDPO  | 5000           | 65.48   | 88.70    | 82.01    | 84.53     | 63.12   | 59.42   | 61.04    | 56.98     | 56.17     | 56.29      | 60.08         | 60.08         | 60.08          |
> > | w/ GeoDPO  | 2500           | 63.95   | 85.32    | 79.68    | 82.09     | 61.89   | 56.21   | 59.23    | 57.22     | 54.42     | 56.87      | 58.73         | 55.99         | 57.61          |
> >
> > We believe these experiments will provide a clearer understanding of the relative contributions of the translator and the reinforcement learning framework in our approach.

---

> > > ### Author Response · Authors · 2025-11-24
> > >
> > > > Q4. Performance when integrated into pipelines like GeoX.
> > >
> > >
> > > We thank the reviewer for this insightful suggestion. We agree that integrating GeoDPO into a complete geometric problem-solving pipeline like GeoX represents a highly promising direction for future research.
> > >
> > > Theoretically, such an integration addresses the critical "perception bottleneck" inherent in current solvers. While systems like GeoX employ a "Geo-ViT" encoder and "GS-Former" to align visual features with formal descriptions, they remain susceptible to distribution shifts and visual hallucinations (e.g., misidentifying a secant as a tangent). By incorporating GeoDPO's translator-guided reinforcement learning framework, the visual encoder can be explicitly optimized to ground geometric primitives with high precision. This ensures that the downstream "formalized program sequences" (which the symbolic solver relies upon) are derived from accurate visual premises, effectively mitigating "garbage-in, garbage-out" errors and enhancing the robustness of the symbolic reasoning engine.
> > >
> > > Regarding empirical validation, we attempted to integrate our method with the open-source GeoX pipeline. However, due to reproducibility issues within the current codebase (specifically, unresolved bugs documented in the repository's issue section, that we also encountered), we were unable to generate immediate quantitative results for this rebuttal. Nevertheless, we view this integration as a logical next step to advance the field and are committed to exploring this direction in our future work.
> > >
> > > **References**
> > >
> > > [1] [Slow Perception: Let's Perceive Geometric Figures Step-by-step](https://arxiv.org/abs/2412.20631)
> > >
> > > [2] [EAGLE: Elevating Geometric Reasoning through LLM-empowered Visual Instruction Tuning](https://arxiv.org/abs/2408.11397)
> > >
> > > [3] [GeoX: Geometric Problem Solving Through Unified Formalized Vision-Language Pre-training](https://arxiv.org/abs/2412.11863)

---

### Official Review · Reviewer_3WG9 · 2025-10-28

**Soundness:** 3
**Presentation:** 2
**Contribution:** 3
**Rating:** 6
**Confidence:** 3

**Summary:**

This paper presents two key components, GeoPerceive and GeoDPO, aimed at enhancing geometric perception in vision-language models. Specifically, the authors propose:
1. GeoPerceive, a novel benchmark that explicitly disentangles geometric perception from reasoning. It employs a canonical domain-specific language (GeoDSL) to represent diagrams in a precise and unambiguous manner.
2. GeoDPO, a translator-guided reinforcement learning framework that improves perception alignment by leveraging natural-language-to-DSL translation as a fine-grained reward signal, while maintaining consistency with the model’s natural-language pretraining distribution.

The authors conduct a relatively comprehensive set of experiments to validate the effectiveness of their proposed methods.

**Strengths:**

1. The authors clearly identify geometric perception as a distinct subproblem within vision-language reasoning, separating low-level perception (recognition of geometric primitives and relations) from high-level symbolic reasoning (logical deduction). This represents a novel and well-motivated decomposition that has not been explicitly formalized in prior VLM research.
2. The paper conducts extensive benchmarking across multiple major vision-language model backbones (Qwen2.5-VL, InternVL3, LLaVA-Next). The proposed method achieves substantial quantitative gains, and downstream reasoning tasks gains. Detailed element-level F1 analyses (covering points, lines, circles, and constraints) further confirm consistent improvements across all geometric subtypes.
3. The GeoDPO formulation is mathematically well-specified, grounded in a principled DPO objective augmented by a translator-based reward model. The authors carefully address critical challenges such as permutation invariance, distributional shift, and the design of fine-grained reward signals, ensuring both theoretical rigor and practical stability.
4. The implementation and codebase are open-sourced, providing transparency and reproducibility for future research.

**Weaknesses:**

1. In Section 3, the paper introduces the GeoPerceive Benchmark, but it does not provide sufficient details regarding its categorical composition, dataset scale, or comparative positioning relative to existing benchmarks. It is recommended that the authors include a summary table outlining the dataset’s structure, sample counts, and distinctions from related benchmarks to improve clarity and reproducibility.
2. The experimental comparisons primarily focus on SFT-based fine-tuning, without incorporating other reinforcement or reward-based baselines (e.g., PPO, DAPO, or similar methods) that could leverage scalar rewards derived from GeoDSL F1 scores. Including such baselines would strengthen the empirical validation of GeoDPO’s effectiveness and robustness.
3. The qualitative analysis of perception failures remains limited. Readers currently lack a clear understanding of the specific types of perception errors that GeoDPO mitigates. It is recommended to include before–after visualization pairs illustrating vision-language model outputs on representative examples, both before and after GeoDPO training, to better demonstrate qualitative improvements.

**Questions:**

1. Do the authors anticipate that GeoDPO could generalize to 3D geometric perception or other structured visual domains such as charts, circuit diagrams, or mechanical drawings? A discussion on this potential extension would strengthen the paper’s generalizability claims and highlight broader applicability.
2. Did the authors attempt to apply PPO or DAPO using the same reward signal? If not, it would be valuable to discuss the expected behavioral differences, computational trade-offs, and design rationale behind choosing DPO over other preference-optimization methods. Clarifying this choice would help readers better understand the methodological positioning of GeoDPO within the broader landscape of reinforcement and alignment approaches.
3. Regarding the downstream performance improvements, it would be helpful if the authors could provide additional results on benchmarks beyond MathVista, to demonstrate the model’s transferability and robustness across diverse geometric reasoning tasks.

---

> ### Author Response · Authors · 2025-11-24
>
> We sincerely thank you for your thoughtful and constructive feedback. We are grateful for your recognition of our paper's distinct problem formulation, methodological rigor, and the substantial quantitative gains achieved by GeoDPO. We particularly appreciate your suggestion to expand our baselines and clarify the benchmark details, which has significantly strengthened our manuscript. Below, we address your comments point-by-point.
>
> > W1. Clarity of GeoPerceive Benchmark Details.
>
> Thank you for pointing this out. We agree that a consolidated summary and comparison in the main text is essential for clarity.
>
> We have revised the manuscript to provide better visibility into the dataset structure. Detailed statistics for the training and testing sets are provided in **Appendix D** of the original version, and we have added a new **Table 9** in **Appendix D** that explicitly compares GeoPerceive with existing benchmarks (Geometry3K, UniGeo, MathVista, GeoQA). This table highlights GeoPerceive's unique positioning: unlike reasoning-focused benchmarks that rely on binary answers and finite samples, GeoPerceive focuses on perception using a canonical, ambiguity-free DSL with support for infinite automated data generation.
>
> > W2 & Q2. Regarding Other RL Baselines
>
> Thank you for the suggestions. We would like to clarify that DPO is not a competing approach, but rather an implementation choice within our proposed RL framework. DPO was selected due to its simplicity and direct applicability to the problem at hand. To address your concern, we conduct additional experiments using PPO, which involves training both a value network and a policy network. We find that PPO is generally slower due to the need for an additional value model, but it still performs on par with DPO in our framework (as shown in the following table). We believe this demonstrates the flexibility of our framework in leveraging different RL implementations.
>
> | Method     | Overall | Points_P | Points_R | Points_F1 | Lines_P | Lines_R | Lines_F1 | Circles_P | Circles_R | Circles_F1 | Constraints_P | Constraints_R | Constraints_F1 |
> | ---------- | ------- | -------- | -------- | --------- | ------- | ------- | -------- | --------- | --------- | ---------- | ------------- | ------------- | -------------- |
> | Pretrained | 57.96   | 84.58    | 75.19    | 79.07     | 64.15   | 49.73   | 53.95    | 45.16     | 44.82     | 44.63      | 54.30         | 54.12         | 54.18          |
> | w/ SFT     | 64.02   | 83.81    | 82.18    | 82.21     | 64.49   | 55.74   | 58.02    | 58.07     | 57.61     | 57.56      | 58.05         | 58.76         | 58.29          |
> | w/ GeoDPO  | 66.19   | 93.97    | 85.96    | 89.26     | 68.82   | 60.65   | 62.30    | 49.38     | 48.54     | 48.70      | 64.50         | 64.50         | 64.50          |
> | w/ GeoPPO  | 65.54   | 91.13    | 86.18    | 87.89     | 69.05   | 59.81   | 61.58    | 56.51     | 55.70     | 55.66      | 57.12         | 56.98         | 57.04          |
>
>
>
> > W3. Qualitative Analysis
>
> We agree that visualizing the specific improvements is crucial. We have added a new section, **Appendix E.3**, which includes **Figure 6** and a detailed case study. This comparison illustrates a representative example where the pretrained model suffers from severe hallucinations (e.g., misidentifying chords and failing to perceive cyclic quadrilaterals), while the GeoDPO-trained model accurately grounds the majority of visual elements.
>
> > Q1. Generalizability to 3D and Other Domains
>
> We strongly anticipate that the GeoDPO framework is highly generalizable to 3D geometry and other structured visual domains, such as charts and circuit diagrams, because its core mechanism (i.e., aligning natural language with a domain-specific reward signal) is inherently domain-agnostic.
>
> By defining a suitable DSL and employing an automated data engine to train a domain-specific translator, this framework can provide fine-grained, perception-focused reward signals to effectively mitigate hallucinations in any new domain without requiring human-labeled data.

---

> > ### Author Response · Authors · 2025-11-24
> >
> > > Q3. Downstream Performance Beyond MathVista
> >
> > We appreciate your suggestion to provide additional results on benchmarks beyond MathVista to demonstrate the model’s transferability and robustness across diverse geometric reasoning tasks.
> >
> > To further demonstrate the transferability and robustness of our model, we evaluated GeoDPO on four additional downstream geometric reasoning benchmarks: GeoQA, Geometry3K, UniGeo (calculation split), and PGPS9k.
> > As shown in the table below, GeoDPO consistently outperforms the SFT baseline across all datasets. This indicates that the enhanced geometric perception gained through our method effectively transfers to broader reasoning tasks, likely because accurate grounding is a prerequisite for correct symbolic reasoning.
> >
> >
> > | Model \ Datasets | GeoQA | Geometry3K | UniGeo(calc) | PGPS9k |
> > | ---------------- | ----- | ---------- | ------------ | ------ |
> > | Pretrained       | 68.1  | 57.6       | 64.3         | 53.7   |
> > | w/ SFT           | 67.6  | 52.3       | 65.1         | 52.5   |
> > | w/ GeoDPO        | 70.2  | 61.4       | 65.5         | 55.7   |

---

> > > ### Comment · Reviewer_3WG9 · 2025-11-28
> > >
> > > The author has addressed most of my concerns, and I am inclined to maintain my positive score.

---

### Official Review · Reviewer_cHmt · 2025-10-31

**Soundness:** 2
**Presentation:** 3
**Contribution:** 2
**Rating:** 6
**Confidence:** 4

**Summary:**

1. This paper proposes GEODSL to uniquely describe each specific diagram, and builds GEOPERCEIVE based on it, combining a generation engine and a solving engine to randomly generate DSL-diagram pairs. It provides low-cost and high-quality data for training and evaluating geometric perception models;
2. They trains a NL-to-DSL translator based on GEOPERCEIVE data, and combines it with the scoring metric to form an effective reward model. It solves the difficulty of scoring caused by the equivalence of order-permutation or shift of natural language description, and makes the training for VLM remain in the original NL field;
3. Experiments on Qwen2.5-VL, InternVL3, and LLaVA Next show that, compared with raw or SFT models, DPO training based on GEOPERCEIVE and the translator can effectively improve the perception ability, as well as achieve improvement in downstream reasoning tasks.

**Strengths:**

1. This paper provides a rigorous and clear exposition of its methodology and experiment;
2. The proposed GEOPERCEIVE pipeline provides an efficient automation scheme for the generation of geometric perception related data;
3. It emphasizes the distinction between perception and reasoning, realizes the transformation from ambiguous natural language to strict DSL through NL-to-DSL translator, and provides a standard reward model for geometric perception results.

**Weaknesses:**

1. Lack of comparison with related methods in the experiment (e.g. [Slow Perception](https://arxiv.org/abs/2412.20631), [EAGLE](https://arxiv.org/abs/2408.11397));
2. Partial experimental results show no significant improvement compared to the SFT method.

**Questions:**

1. How do you view the lag of DPO training compared to SFT in some specific indicators on the main-test split? (For example, the F1 score of circles perception, which serves as the aspect with the lowest score among all raw models)
2. Does the GEODPO perform better than other methods, such as [Slow Perception](https://arxiv.org/abs/2412.20631) and [EAGLE](https://arxiv.org/abs/2408.11397)?

---

> ### Author Response · Authors · 2025-11-24
>
> We are grateful for your detailed and constructive feedback. We are pleased to hear that the methodology, experiment exposition, and the GeoPerceive pipeline have been recognized as valuable contributions. Below, we address each of the points raised, with clarifications and additional experimental insights.
>
> > W1 & Q2. Comparison with Related Methods. (e.g., Slow Perception[1], EAGLE[2])
>
> We appreciate your suggestion to compare our approach with related methods like Slow Perception and EAGLE. These comparisons are indeed valuable for providing broader context and highlighting the relative strengths of our approach.
>
> **1) Comparison with Slow Perception.**
>
> In the geometric reasoning task, we follow the approach from Slow Perception, using a geometric parsing module as an external component to enhance the reasoning capabilities of GPT-4o. In our setup, GeoDPO serves as the perception module. The results show that GeoDPO significantly outperforms Slow Perception, achieving a MathVista score of 63.6, compared to 60.1 for the latter.
>
> | Model                   | MathVista |
> | ----------------------- | ----- |
> | gpt-4o                  | 53.4  |
> | gpt-4o + SlowPerception | 60.1  |
> | gpt-4o + GeoDPO         | 63.6  |
>
> **2) Comparison with EAGLE.**
>
> EAGLE presents some interesting approaches to geometric perception. However, as it is not open-source, we are unable to directly access or evaluate its model. Based on publicly available results from EAGLE’s reported benchmarks, our method demonstrates superior performance on reasoning tasks such as the GeoQA task. Our model achieves a score of 70.2, surpassing EAGLE’s 67.1. This clearly highlights the effectiveness and advantage of GeoDPO in geometric reasoning.
>
> | Model                   | GeoQA |
> | ----------------------- | ----- |
> | EAGLE                   | 67.1  |
> | GeoDPO                  | 70.2  |
>
>
> > W2 & Q1. Performance and Lag of DPO Training Compared to SFT.
>
> This is a very astute observation, and we thank you for pointing this out. Indeed, for certain model-category pairs, such as the **Qwen2.5-VL** model’s performance on **Circles perception**, SFT does outperform GeoDPO.
>
> We interpret this discrepancy not as a failure of GeoDPO but as indicative of a **trade-off between in-domain specialization and robust generalization**:
>
> 1. **SFT Overfits, GeoDPO Generalizes.** SFT fine-tunes the model to closely fit patterns observed in the training data. While this can result in high in-domain performance, it often leads to overfitting, particularly in tasks where certain structures (e.g., circles) are overrepresented. As a result, the model may score highly on familiar patterns but struggle to generalize beyond them.
> 2. **GeoDPO's Limitation from Reward Signal.** In contrast, GeoDPO’s performance depends on the quality of the reward signal derived from the NL-to-DSL translator. As shown in **Table 2**, the translator performs significantly worse on "Circles" (74.6 F1) than on "Points" (96.1 F1) or "Lines" (95.2 F1). This indicates that GeoDPO’s comparatively lower performance on Circles is a reflection of noisier and less reliable reward signals, rather than a failure of the approach itself.
> 3. **OOD Test Confirms This.** Notably, when evaluated on out-of-distribution (OOD) data (see **Table 4**), the in-domain advantage of SFT on Circles disappears. SFT's Circle F1 drops from 57.56 to 45.92, whereas GeoDPO maintains a comparable score, highlighting its superior ability to generalize. Further, on the InternVL benchmark, SFT’s Circle F1 drops to 49.02 (a −4.03% decline), while GeoDPO improves to 52.17 (a +3.35% gain), further supporting this claim.
>
> In summary, while SFT excels at memorizing domain-specific patterns, GeoDPO demonstrates stronger **generalization ability**, especially in OOD scenarios. This supports our central claim that GeoDPO provides a more robust and adaptable solution for geometric perception and reasoning.
>
> **References**
>
> [1] [Slow Perception: Let's Perceive Geometric Figures Step-by-step](https://arxiv.org/abs/2412.20631)
>
> [2] [EAGLE: Elevating Geometric Reasoning through LLM-empowered Visual Instruction Tuning](https://arxiv.org/abs/2408.11397)

---

### Official Review · Reviewer_MwSA · 2025-11-01

**Soundness:** 2
**Presentation:** 3
**Contribution:** 2
**Rating:** 2
**Confidence:** 5

**Summary:**

This paper presents GEOPERCEIVE, a new benchmark and synthetic data pipeline for evaluating and training geometric perception in vision-language models (VLMs) using diagrams paired with a canonical domain-specific language (GeoDSL). To improve VLMs' geometric grounding, the authors propose GEODPO, a translator-guided reinforcement learning (RL) framework that uses an NL-to-DSL translator to provide fine-grained reward signals for DPO-based preference optimization. Extensive experiments on both in-domain and out-of-distribution setups show that GEODPO yields stronger and more robust improvements over supervised fine-tuning (SFT), especially in geometric perception and downstream reasoning tasks

**Strengths:**

1. Benchmark and Synthetic Data Pipeline: The paper introduces GEOPERCEIVE, which systematically generates and renders diagrams with unambiguous geometric DSLs. The pipeline creates diverse complexity-controlled data at scale, providing a solid resource for both evaluation and training.

2. Meaningful Visualizations: Figures such as Figure 1 (exposing ambiguity in existing DSLs), Figure 2 (illustrating GeoDSL syntax), Figure 3 (full pipeline including RL steps), and Figure 4 (showing data complexity across iterations), are thoughtfully constructed. Figure 3, in particular, makes the RL and preference generation workflow concrete, aiding reproducibility and interpretability

**Weaknesses:**

1. Lack of Rigor in the Task: The geometric perception task is not merely a simple perceptual task. Specifically, the geometric information obtained through basic image perception should undergo rigorous verification (at least ensuring that the perceived constraints are consistent and non-conflicting). For example, certain relationships in the image, such as perpendicularity or numerical data, must be strictly validated against predefined conditions.

2. Insufficient Explanation of Geometric Relations: In the defined quadruplet (comprising points, lines, and circles), can these elements sufficiently cover all basic geometric entities? Additionally, what specific constraints are included? Further clarification is needed to demonstrate the rationality and completeness of the definition.

3. Regarding OOD Data: Further clarification is needed on the distinction between Out-of-Distribution (OOD) data and the benchmark data. Specifically, why is the manually annotated data considered out-of-distribution? Is this due to differences in drawing style, or does it pertain more to the complexity and nature of the geometric relationships?

4. Limited Analysis of Failure Cases and Limitations: The main text barely addresses where GEODPO (or GeoDSL itself) struggles. Section E (Limitations) is confined to algebraic constraints, but a frank, quantitative error analysis (e.g., visualizations of failed cases, confusion matrices on specific primitives or constraints, qualitative outlier examples) is absent. For instance, Figure 4 shows only successful, rendered samples; missing are diagrams or programs leading to failure, which could inform limitations of the framework or concrete bottlenecks in the translation pipeline or RL optimization.

5. Ablation and Baseline Completeness: The experimental section (Table 3, Table 4) lacks comparison to several baseline RL/population-based approaches used in the field, including those explicitly referenced in missing related works. There is also a missed opportunity for ablation—e.g., separating out the effect of NL2DSL translator quality from the DPO RL signal, or evaluating against other forms of preference optimization.

**Questions:**

please see the weaknesses

---

> ### Author Response · Authors · 2025-11-24
>
> We deeply appreciate your thoughtful feedback on our paper and the recognition of our contributions in terms of the novel GeoPerceive pipeline and GeoDPO framework. Below, we address the specific points raised and provide clarifications and additional experiments to further strengthen our claims.
>
> > W1. Rigor in the Task
>
> Thank you for pointing out the concern regarding the rigor of the geometric perception task. We want to clarify that our approach is indeed rigorous, with several safeguards in place to ensure consistency and validity:
>
> 1. **Regarding Explicit Constraint Handling.** In **Section 3.1**, we formally define the diagram representation as \$G = \langle \mathcal{P}, \mathcal{L}, \mathcal{C}, \mathcal{R} \rangle\$, where \$\mathcal{P}, \mathcal{L}, \mathcal{C}\$ represent the points, lines, and circles, and \$\mathcal{R}\$ refers to geometric constraints such as perpendicularity and tangency. The formal definitions in **Appendix A** also highlight the strict and well-defined relations we use for constraints.
>
> 2. **Regarding Data Generation.** Our solving engine (described in **Section 3.4** and **Appendix C**) is designed to ensure geometric consistency. We employ loss functions based on geometric constraints to verify the validity of each rendered diagram. This approach ensures that the generated diagrams are geometrically sound. Discarded samples, which are unsolved, are shown in **Figure 4**, while the retained samples, which meet the geometric requirements, are shown in **Figure 5**. This demonstrates that the retained diagrams are geometrically valid, effectively preventing unsolvable or low-quality diagrams from being included in the training process.
>
> 3. **Regarding Evaluation Metrics.** We employ a weighted F1-score (**Equation 3**) in our evaluation, which directly incorporates the constraints as part of the score computation. This metric guarantees that the perception model is evaluated not only on the geometric primitives but also on its ability to handle constraints in a geometrically consistent manner.
>
> Thus, our method is both rigorous and precise in measuring geometric perception.
>
> > W2. Clarification of Geometric Relations.
>
> We appreciate the suggestion to elaborate on the sufficiency of the geometric relations defined in our model.
>
> 1. **Regarding Primitives.** In our work, the geometric primitives (including points, lines, and circles) are sufficient to represent a broad class of geometric constructions, especially in the context of Euclidean geometry. As demonstrated in related works such as Geometry3K[1] and GeoModelBuilder[2], these three primitives are adequate for handling most of the Euclidean geometry problems without the need for more complex algebraic curves.
>
> 2. **Regarding Constraints.**: The full list of constraints we consider is provided in **Table 7 (Appendix A)**. These include fundamental relationships such as perpendicularity, tangency, and parallelism. Together, these constraints form a comprehensive set of standard geometric relations, enabling us to assess the model's performance in a thorough and holistic manner.
>
> We believe these choices strike a balance between generality and specificity, ensuring that the framework can be used to evaluate a wide variety of geometric perception problems.
>
> > W3. Regarding OOD Data
>
> We recognize the importance of clarifying the distinction between in-domain and out-of-domain (OOD) data. The motivation behind the GeoPerceive-OOD dataset is to test the generalization capabilities of our model.
>
> Unlike in-domain data, which is generated through our pipeline, OOD data comes from real-world sources, with annotations manually created by expert annotators as described in **Section 5.3.2**. These diagrams differ significantly in terms of drawing styles, complexity, and the variety of geometric relations presented. This diversity is crucial to assessing whether our model can generalize to scenarios beyond the scope of our synthetic training data.
>
> We hope this clarifies the rationale behind our use of OOD data.
>
> > W4. Analysis of Failure Cases and Limitations
>
> We appreciate your feedback on the limited analysis of failure cases. While **Section D** does include a quantitative analysis of the solving engine’s success rate, we agree that failure cases and outlier examples are important for fully understanding the limitations of the framework.
>
> Therefore, we have added additional visualizations and error analyses of failure cases in **Appendix C.6**. This includes diagrams where the solver fails to converge due to conflicting constraints or overly complex geometric configurations.
>
> We believe that these additions will provide a more comprehensive understanding of the limitations and help to contextualize the results.

---

> ### Author Response · Authors · 2025-11-24
>
> > W5. Ablation and Baseline Completeness
>
> We understand your concerns regarding the comparison with baseline approaches and the need for a more comprehensive ablation study.
>
> **1) PPO vs DPO.**
>
> Thank you for the suggestions. We would like to clarify that DPO is not a competing approach, but rather an implementation choice within our proposed RL framework. DPO was selected due to its simplicity and direct applicability to the problem at hand. To address your concern, we conduct additional experiments using PPO, which involves training both a value network and a policy network. We find that PPO is generally slower due to the need for an additional value model, but it still performs on par with DPO in our framework (as shown in **Table R1**). We believe this demonstrates the flexibility of our framework in leveraging different RL implementations.
>
> > **Table R1. Comparison of PPO and DPO**
> | Method     | Overall | Points_P | Points_R | Points_F1 | Lines_P | Lines_R | Lines_F1 | Circles_P | Circles_R | Circles_F1 | Constraints_P | Constraints_R | Constraints_F1 |
> | ---------- | ------- | -------- | -------- | --------- | ------- | ------- | -------- | --------- | --------- | ---------- | ------------- | ------------- | -------------- |
> | Pretrained | 57.96   | 84.58    | 75.19    | 79.07     | 64.15   | 49.73   | 53.95    | 45.16     | 44.82     | 44.63      | 54.30         | 54.12         | 54.18          |
> | w/ SFT     | 64.02   | 83.81    | 82.18    | 82.21     | 64.49   | 55.74   | 58.02    | 58.07     | 57.61     | 57.56      | 58.05         | 58.76         | 58.29          |
> | w/ GeoDPO  | 66.19   | 93.97    | 85.96    | 89.26     | 68.82   | 60.65   | 62.30    | 49.38     | 48.54     | 48.70      | 64.50         | 64.50         | 64.50          |
> | w/ GeoPPO  | 65.54   | 91.13    | 86.18    | 87.89     | 69.05   | 59.81   | 61.58    | 56.51     | 55.70     | 55.66      | 57.12         | 56.98         | 57.04          |

---

> ### Author Response · Authors · 2025-11-24
>
> **2) Translator Quality.**
>
> To isolate the effect of the NL2DSL translator from the DPO reinforcement learning signal, we conduct an ablation study in which we train a weaker translator, using less data, and thus achieving lower performance. As shown in **Table R2** and **Table R3**, the performance of the GeoDPO framework significantly degrades when the quality of the translator is compromised. This confirms that the quality of the translator directly influences the effectiveness of the reward signal in our model.
>
> > **Table R2. Performance of Translator Trained on Different Amounts of Data.**
> | Data        | Validity | Overall | Points_P | Points_R | Points_F1 | Lines_P | Lines_R | Lines_F1 | Circles_P | Circles_R | Circles_F1 | Constraints_P | Constraints_R | Constraints_F1 |
> | ----------- | -------- | ------- | -------- | -------- | --------- | ------- | ------- | -------- | --------- | --------- | ---------- | ------------- | ------------- | -------------- |
> | 9800 (ours) | 100.0    | 89.2    | 96.4     | 95.9     | 96.1      | 95.4    | 95.0    | 95.2     | 75.0      | 74.8      | 74.6       | 91.7          | 90.4          | 90.8           |
> | 7500        | 99.5     | 89.1    | 94.9     | 94.7     | 94.8      | 94.3    | 94.2    | 94.2     | 76.7      | 75.9      | 76.2       | 91.9          | 91.0          | 91.3           |
> | 5000        | 97.5     | 82.3    | 94.5     | 86.8     | 93.7      | 91.7    | 91.6    | 90.1     | 73.9      | 73.1      | 60.4       | 71.6          | 70.3          | 85.0           |
> | 2500        | 95.5     | 73.8    | 92.9     | 80.4     | 93.7      | 90.5    | 83.2    | 84.6     | 66.0      | 65.6      | 56.7       | 57.5          | 57.3          | 60.3           |
>
> > **Table R3. Performance of GeoDPO with Rewards from Translators of Varying Quality.**
> | Method     | TranslatorData | Overall | Points_P | Points_R | Points_F1 | Lines_P | Lines_R | Lines_F1 | Circles_P | Circles_R | Circles_F1 | Constraints_P | Constraints_R | Constraints_F1 |
> | ---------- | -------------- | ------- | -------- | -------- | --------- | ------- | ------- | -------- | --------- | --------- | ---------- | ------------- | ------------- | -------------- |
> | Pretrained | N/A            | 57.96   | 84.58    | 75.19    | 79.07     | 64.15   | 49.73   | 53.95    | 45.16     | 44.82     | 44.63      | 54.30         | 54.12         | 54.18          |
> | w/ GeoDPO  | 9800 (ours)    | 66.19   | 93.97    | 85.96    | 89.26     | 68.82   | 60.65   | 62.30    | 49.38     | 48.54     | 48.70      | 64.50         | 64.50         | 64.50          |
> | w/ GeoDPO  | 7500           | 66.27   | 93.08    | 84.48    | 86.61     | 62.39   | 60.01   | 61.35    | 58.71     | 56.77     | 57.67      | 60.15         | 58.94         | 59.45          |
> | w/ GeoDPO  | 5000           | 65.48   | 88.70    | 82.01    | 84.53     | 63.12   | 59.42   | 61.04    | 56.98     | 56.17     | 56.29      | 60.08         | 60.08         | 60.08          |
> | w/ GeoDPO  | 2500           | 63.95   | 85.32    | 79.68    | 82.09     | 61.89   | 56.21   | 59.23    | 57.22     | 54.42     | 56.87      | 58.73         | 55.99         | 57.61          |
>
>
> We believe these experiments will provide a clearer understanding of the relative contributions of the translator and the reinforcement learning framework in our approach.
>
>
> **References**
>
> [1] [Inter-GPS: Interpretable Geometry Problem Solving with Formal Language  and Symbolic Reasoning](https://arxiv.org/abs/2105.04165)
>
> [2] [Automatically Building Diagrams for Olympiad Geometry Problems](https://arxiv.org/abs/2012.02590)

---

### Author Response · Authors · 2025-12-01
**General Response (1/2)**

We thank the Area Chair and all reviewers for your invaluable time and constructive feedback. We are encouraged that the reviewers recognize:
1. The **novelty** and **rigor** of our problem formulation, which explicitly disentangles perception from reasoning (cHmt, 3WG9, nm2u);
2. The **scalability** and **automation** of our **GeoPerceive** pipeline (MwSA, cHmt, nm2u);
3. The **effectiveness** of the **GeoDPO** framework for RL (3WG9, nm2u);
4. The **extensiveness** of our experimental evaluation (3WG9);
5. The **reproducibility** of our work (3WG9);
6. The **meaningful visualizations** that aid interpretability (MwSA).


In this rebuttal, we have carefully addressed all individual concerns and clarified misunderstandings. Specifically, we have conducted **extensive additional experiments** to strengthen our empirical validation, including:
1.  **RL Baselines:** Comparison with PPO to validate our choice of DPO.
2.  **SOTA Comparisons:** Comparison with SlowPerception, EAGLE, and GeoX.
3.  **Expanded Benchmarks:** Evaluation on 4 additional datasets (GeoQA, Geometry3K, UniGeo, PGPS9k).
4.  **Ablation Studies:** Analysis of the translator quality's impact on reward signals.
5.  **Qualitative Analysis:** Visualization of failure cases and perceptual improvements.

Below, we summarize our responses to the common concerns raised by multiple reviewers.

> **1. Comparison with Other RL Methods (PPO vs. DPO) [MwSA, 3WG9, nm2u]**

We would like to clarify that DPO is not a competing approach, but rather an implementation choice within our proposed RL framework. DPO was selected due to its simplicity and direct applicability to the problem at hand. To address your concern, we conduct additional experiments using PPO, which involves training both a value network and a policy network. We find that PPO is generally slower due to the need for an additional value model, but it still performs on par with DPO in our framework (as shown in the following table). We believe this demonstrates the flexibility of our framework in leveraging different RL implementations.

> **Table R1. Comparison of PPO and DPO**
| Method     | Overall | Points_P | Points_R | Points_F1 | Lines_P | Lines_R | Lines_F1 | Circles_P | Circles_R | Circles_F1 | Constraints_P | Constraints_R | Constraints_F1 |
| ---------- | ------- | -------- | -------- | --------- | ------- | ------- | -------- | --------- | --------- | ---------- | ------------- | ------------- | -------------- |
| Pretrained | 57.96   | 84.58    | 75.19    | 79.07     | 64.15   | 49.73   | 53.95    | 45.16     | 44.82     | 44.63      | 54.30         | 54.12         | 54.18          |
| w/ SFT     | 64.02   | 83.81    | 82.18    | 82.21     | 64.49   | 55.74   | 58.02    | 58.07     | 57.61     | 57.56      | 58.05         | 58.76         | 58.29          |
| w/ GeoDPO  | 66.19   | 93.97    | 85.96    | 89.26     | 68.82   | 60.65   | 62.30    | 49.38     | 48.54     | 48.70      | 64.50         | 64.50         | 64.50          |
| w/ GeoPPO  | 65.54   | 91.13    | 86.18    | 87.89     | 69.05   | 59.81   | 61.58    | 56.51     | 55.70     | 55.66      | 57.12         | 56.98         | 57.04          |

---

> ### Author Response · Authors · 2025-12-01
> **General Response (2/2)**
>
> > **2. Performance on Downstream Reasoning [3WG9, nm2u]**
>
> To further demonstrate the transferability and robustness of our model, we evaluated GeoDPO on four additional downstream geometric reasoning benchmarks: GeoQA, Geometry3K, UniGeo (calculation split), and PGPS9k. As shown in the table below, GeoDPO consistently outperforms the SFT baseline across all datasets. This indicates that the enhanced geometric perception gained through our method effectively transfers to broader reasoning tasks, likely because accurate grounding is a prerequisite for correct symbolic reasoning.
>
> > **Table GR2. Extensive Downstream Evaluation.**
> | Model \ Datasets | GeoQA | Geometry3K | UniGeo(calc) | PGPS9k |
> | ---------------- | ----- | ---------- | ------------ | ------ |
> | Pretrained       | 68.1  | 57.6       | 64.3         | 53.7   |
> | w/ SFT           | 67.6  | 52.3       | 65.1         | 52.5   |
> | w/ GeoDPO        | 70.2  | 61.4       | 65.5         | 55.7   |
>
> > **3. Impact of Translator Quality [MwSA, nm2u]**
>
> To isolate the effect of the NL2DSL translator from the DPO reinforcement learning signal, we conduct an ablation study in which we train a weaker translator, using less data, and thus achieving lower performance. As shown in the following tables, the performance of the GeoDPO framework significantly degrades when the quality of the translator is compromised. This confirms that the quality of the translator directly influences the effectiveness of the reward signal in our model.
>
> > **Table GR3. Performance of Translator Trained on Different Amounts of Data.**
> | Data        | Validity | Overall | Points_P | Points_R | Points_F1 | Lines_P | Lines_R | Lines_F1 | Circles_P | Circles_R | Circles_F1 | Constraints_P | Constraints_R | Constraints_F1 |
> | ----------- | -------- | ------- | -------- | -------- | --------- | ------- | ------- | -------- | --------- | --------- | ---------- | ------------- | ------------- | -------------- |
> | 9800 (ours) | 100.0    | 89.2    | 96.4     | 95.9     | 96.1      | 95.4    | 95.0    | 95.2     | 75.0      | 74.8      | 74.6       | 91.7          | 90.4          | 90.8           |
> | 7500        | 99.5     | 89.1    | 94.9     | 94.7     | 94.8      | 94.3    | 94.2    | 94.2     | 76.7      | 75.9      | 76.2       | 91.9          | 91.0          | 91.3           |
> | 5000        | 97.5     | 82.3    | 94.5     | 86.8     | 93.7      | 91.7    | 91.6    | 90.1     | 73.9      | 73.1      | 60.4       | 71.6          | 70.3          | 85.0           |
> | 2500        | 95.5     | 73.8    | 92.9     | 80.4     | 93.7      | 90.5    | 83.2    | 84.6     | 66.0      | 65.6      | 56.7       | 57.5          | 57.3          | 60.3           |
>
> > **Table GR4. Performance of GeoDPO with Rewards from Translators of Varying Quality.**
> | Method     | TranslatorData | Overall | Points_P | Points_R | Points_F1 | Lines_P | Lines_R | Lines_F1 | Circles_P | Circles_R | Circles_F1 | Constraints_P | Constraints_R | Constraints_F1 |
> | ---------- | -------------- | ------- | -------- | -------- | --------- | ------- | ------- | -------- | --------- | --------- | ---------- | ------------- | ------------- | -------------- |
> | Pretrained | N/A            | 57.96   | 84.58    | 75.19    | 79.07     | 64.15   | 49.73   | 53.95    | 45.16     | 44.82     | 44.63      | 54.30         | 54.12         | 54.18          |
> | w/ GeoDPO  | 9800 (ours)    | 66.19   | 93.97    | 85.96    | 89.26     | 68.82   | 60.65   | 62.30    | 49.38     | 48.54     | 48.70      | 64.50         | 64.50         | 64.50          |
> | w/ GeoDPO  | 7500           | 66.27   | 93.08    | 84.48    | 86.61     | 62.39   | 60.01   | 61.35    | 58.71     | 56.77     | 57.67      | 60.15         | 58.94         | 59.45          |
> | w/ GeoDPO  | 5000           | 65.48   | 88.70    | 82.01    | 84.53     | 63.12   | 59.42   | 61.04    | 56.98     | 56.17     | 56.29      | 60.08         | 60.08         | 60.08          |
> | w/ GeoDPO  | 2500           | 63.95   | 85.32    | 79.68    | 82.09     | 61.89   | 56.21   | 59.23    | 57.22     | 54.42     | 56.87      | 58.73         | 55.99         | 57.61          |
>
> We hope these additional experiments and clarifications satisfactorily address the reviewers' concerns. We are committed to incorporating these results into the final version of the paper.

---

### Meta-Review · Area_Chair_bCAT · 2026-01-07

**Summary:**

All reviewers recognized GeoPerceive + GeoDPO as a well-motivated and technically sound contribution that addresses a genuine bottleneck in vision-language models: the inability to disentangle geometric perception from reasoning.

Initial concerns centered on experimental completeness, baseline coverage, failure analysis depth, and clarity around task rigor and OOD definition.

**Reviewer Concerns:**

The rebuttal added substantial new experiments (PPO baseline, SOTA comparisons, downstream benchmarks, translator ablations, qualitative failures), which addressed part of reviewer criticisms.

**Reviewer Scores:**

Most of the issues are addressed in the rebuttal. Minor points that may persist include:

Whether the framework extends naturally to 3D geometry or other structured visual domains (acknowledged and discussed).

Some limitations of GeoDSL (e.g., algebraic constraints), already noted in the paper.

These are reasonable scope limitations rather than flaws.

---

### Decision · Program_Chairs · 2026-01-26

Accept (Poster)